# Benefits of sea ice initialization for the interannual-to-decadal climate prediction skill in the Arctic in EC-Earth3

Tian Tian[1], Shuting Yang[1], Mehdi Pasha Karami[2], François Massonnet[3], Tim Kruschke[2], and Torben Koenigk[2,4]

[1]Danish Meteorological Institute, Denmark
[2]Swedish Meteorological and Hydrological Institute, Sweden
[3]Georges Lemaitre Centre for Earth and Climate Research, Earth and Life Institute, Université Catholique de Louvain, Louvain-la-Neuve, Belgium
[4]Bolin Centre for Climate Research, Stockholm University, Sweden

**Correspondence:** Tian Tian (tian@dmi.dk)

**Abstract.** A substantial part of Arctic climate predictability at interannual time scales stems from the knowledge of the initial sea ice conditions. Among all the variables characterizing sea ice, sea ice volume, being a product of sea ice concentration (SIC) and thickness (SIT), is the most sensitive parameter for climate change. However, the majority of climate prediction systems are only assimilating the observed SIC due to lack of long-term reliable global observation of SIT. In this study the EC-Earth3 Climate Prediction System with anomaly initialization to ocean, SIC and SIT states is developed. In order to evaluate the regional benefits of specific initialized variables, three sets of retrospective ensemble prediction experiments are performed with different initialization strategies: ocean-only; ocean plus SIC; and ocean plus SIC and SIT initialization. In the Atlantic Arctic, the GIN/Barents Seas are the two most skillful regions in SIC prediction for up to 5-6 lead years with ocean initialization; there are reemerging skills for SIC in the Barents/Kara Seas in lead years 7-9 coincided with improved skills of sea surface temperature (SST), reflecting the impact of SIC initialization on ocean-atmosphere oscillations for interannal-to-decadal time scales. For years 2-9 average, the region with significant skill for SIT is found confined to the central Arctic Ocean, covered by multi-year sea ice (CAO-MYI); winter preconditioning with SIT initialization may increase the skill for September SIC in the eastern Arctic (e.g. Kara/Laptev/E. Siberian Seas) and in turn improve the skill of air surface temperature locally and further expanded over land. SIT initialization outperforms the other initialization methods in improving SIT prediction in the Pacific Arctic (e.g. East Siberian/Beaufort Seas) in the first few lead years. Our results suggest that as the climate warming continues and the central Arctic Ocean might become seasonal ice free in the future, the controlling mechanism for decadal predictability may thus shift from being the sea ice volume playing the major role to a more ocean-related processes.

## 1 Introduction

Summer sea ice in the Arctic Ocean has lost nearly three-quarters of its sea ice volume (SIV) since the 1970's (Kwok, 2018) caused by a reduction of both sea ice extent (SIE) and thickness (SIT). This sea ice melt, inducing ice-albedo feedback, contributes to the larger warming of the atmosphere in the Arctic than the global mean, an effect known as polar amplification

(Wadhams, 2012). Observations suggest that the Arctic has warmed at more than twice the rate of the globe (Holland and Bitz, 2003; Serreze and Barry, 2011; Dai et al., 2019). Moreover, the enhanced sea ice melt and associated transports of freshwater to the south weakens the Atlantic meridional overturning circulation and related poleward heat transport on decadal timescales (Sévellec et al., 2017). Therefore, a realistic representation of Arctic sea ice is an essential element of a coupled climate prediction system.

Persistence has been recognized as a primary source of Arctic sea ice predictability in the last ten years (Blanchard-Wrigglesworth et al., 2011; Chevallier and Salas-Mélia, 2012; Chevallier et al., 2019): total SIV is the most persistent variable (∼4 yr) compared to local SIC (∼1 month) and total SIE (∼1 season); local SIT is the second persistent, ranging from seasons in the marginal ice zone (MIZ) to approximately a year in the CAO. Moreover, the persistence timescale is found to be variable in a climate system due to advection of sea ice anomalies, heat exchange between ocean and atmosphere, and changes in climate forcing, as reviewed by Guemas et al. (2016). For example, the memory of SIE can reemerge beyond its own persistence (e.g. 2 to 5 months) by storing memory in the upper-ocean heat content or SIT (Blanchard-Wrigglesworth et al., 2011), hence initialization of winter SIT anomalies provides some predictive capability for summer SIE with winter preconditioning in several studies (Holland et al., 2011; Chevallier and Salas-Mélia, 2012; Blanchard-Wrigglesworth and Bitz, 2014; Day et al., 2014). As another example, reemergence of sea ice anomalies follows modulation of the upper-ocean heat anomalies: if sea ice retreats anomalously early in spring, more heat is stored in the oceanic upper mixed layer that causes later freeze up (Blanchard-Wrigglesworth et al., 2011). Snow cover can affect sea ice predictability in several competing ways: in early spring, the presence of snow causes the local albedo to be high, hence delaying melt onset. However, when snow melts, it forms pools of water at the surface of sea ice known as melt ponds, with a relatively lower albedo (Schröder et al., 2014).

The Arctic sea ice in the Atlantic sector can be predicted from a few years to a decade, because of the strong role exerted by the ocean heat advected from upstream and converged on the position of the sea ice edge (Yeager et al., 2015; Årthun et al., 2017; Dai et al., 2020). Some studies suggested that sea ice persistence in the central Arctic can be modulated by oscillation of the Arctic atmospheric circulation between predominantly cyclonic and anticyclonic circulation regimes over timescales of 5-7 years (Proshutinsky and Johnson, 1997; Armitage et al., 2020). From observations, a remarkable oscillation in SIE and SIV is featuring a pause or enhanced ice loss at a period of 7 years, corresponding to some prominent modes of internal variability, such as the winter North Atlantic Oscillation (NAO, Bitz et al., 1996; Swart et al., 2015; Gascard et al., 2019).

To date most seasonal prediction systems start from a SIT reanalysis data set (Collow et al., 2015; Dirkson et al., 2017), such as from the Pan-Arctic Ice Ocean Model and Assimilation System (PIOMAS, Schweiger et al., 2011). However, most ocean-sea ice reanalyses including PIOMAS do not explicitly assimilate SIT due to inadequate long-term observations of broad coverage. Chevallier et al. (2017) found that the representation of SIT can differ largely due to different local advective processes between models in the intercomparison of 14 state-of-the-art global ocean reanalyses. On the other hand, even fed with identical reanalysis SIT, there was much less agreement in predicting spatial pattern of SIC across dynamical models in the Arctic coasts than the central Arctic (Blanchard-Wrigglesworth et al., 2017). Such a degradation of forecast skill from the central Arctic to the coasts was found in other idealized multi-model experiments to be associated with advective sea ice processes rather than ice thermodynamics (Tietsche et al., 2014).

At decadal time-scales, some studies indicate that directly assimilating SIT can stimulate long-term forecast drift particularly for the total SIV in the Arctic. Alternatively, anomaly initialization (AI), that does not correct the model climate state but anomalies, may efficiently suppress the drift. However, AI may not necessarily improve prediction skill due to the inconsistency between initialized variables. In comparison of the two initialization methods with the EC-Earth2.3 climate model, Volpi et al. (2017) found that the Arctic SIV drifts towards a stabilized state (i.e. stable biases) with full field initialization (FFI) to sea ice states (SIC and SIT) for lead time longer than 5 years, whereas the drift is substantially reduced by AI using identical reconstructed sea ice states. Thus the prediction skill in air temperature at 2 m (TAS) over the Arctic is improved with AI in comparison with FFI. However, the skill difference is found small in predicting the Arctic sea ice area between the two initialized experiments. Despite the established notion that initial information of SIT is a key source of predictability, assimilating it in operational systems actually remains a challenge.

The Arctic Ocean is changing (Jeffries et al., 2013) with a large loss of MYI and a rapid transition from thick perennial towards thinner seasonal sea ice (Wadhams, 2012; Onarheim et al., 2018; Kwok, 2018). In the Beaufort Gyre (BG), sea ice cover becomes thinner and weaker and is strongly associated with the acceleration of wind-driven ocean currents since 1990s (Armitage et al., 2020). In the Barents Sea (BS), northerly wind anomalies can increase sea ice export from the CAO to the BS and reduce the Atlantic inflow through the BS Opening (Dai et al., 2020). It in turn will alter local TAS via ocean and atmosphere heat exchange. This presents a great challenge to predict decadal changes in Arctic sea ice with predictability varying in different regions at different time-scales (Guemas et al., 2016). To our knowledge, most studies on predicting regional Arctic sea ice conditions focused on timescales from a month to a few years (Bushuk et al., 2019; Cruz-García et al., 2019; Kimmritz et al., 2019). Few studies have comprehensively examined the impacts of ocean and sea ice anomaly initialization states in a climate prediction system with respect to decadal time scales. A recent study addressed this topic by only assimilating SST and hence had a special focus on increased skills by SST in the Arctic MIZ (Dai et al., 2020).

The objective of this study is to investigate the decadal prediction skill of Arctic sea ice in the EC-Earth3 Climate Prediction System with anomaly initialization (EC-Earth3-CPSAI) to ocean, SIC and SIT states. We developed a novel method to constrain local SIV anomalies by initializing both SIC and SIT. As the method is developed as a prototype for our initialization strategy implemented for the Coupled Model Intercomparison Project phase 6 (CMIP6) decadal climate prediction project (DCPP) with EC-Earth3, the present study provides a documentation of the new climate prediction system with anomaly initialization including SIT in a multi-category sea ice model framework. It characterizes the performance with focus on the predictions in the Arctic. Three sets of ensemble hindcast experiments are performed and analyzed to evaluate the benefits of respective initialized variables and to quantify the added skill from SIT initialization.

This paper is structured as follows: Section 2 introduces the climate prediction system EC-Earth3-CPSAI as well as the experimental design; Section 3 examines the two sources of forecast error such as model bias, forecast drift as well as the imprint of initial conditions (ICs) in the first year. Section 4 evaluates and discuss the benefits of specific initialized variables at interannual to decadal scales for different Arctic regions. Section 5 is the summary.

## 2 Model system and experiment design

### 2.1 The EC-Earth3 Climate Prediction System with Anomaly Initialization (EC-Earth3-CPSAI)

The EC-Earth is a state-of-the-art Earth System Model developed by the EC-Earth consortium (Doescher et al., 2020). The core of EC-Earth consists of component models for atmosphere, ocean and sea ice, so-called AOGCM. In this study we use the officially released AOGCM configuration of EC-Earth model for contributions to the CMIP6, EC-Earth3 (release v3.3.1.1). The atmospheric component is the Integrated Forecast System (IFS cycle 36r4) developed by the European Centre for Medium Range Weather Forecasts (ECMWF). It uses the TL255 horizontal grid (i.e. triangular truncation at wavenumber 255 in spectral space with a linearly reduced Gaussian grid, corresponding to a spacing of about 80 km) and 91 vertical model levels with the top level at 0.01 hPa. The ocean component is the Nucleus for European Modelling of the Ocean, version 3.6 (NEMO3.6) coupled to the Louvain-la-Neuve sea Ice Model, version 3 (LIM3, Rousset et al., 2015). NEMO has a family of global ocean tripolar horizontal grids (called ORCA, Madec et al., 2019). The NEMO-LIM3 is configured with a nominal $1°$ resolution horizontal grid (i.e. ORCA1) and 75 vertical levels. It is worth noting that the sea ice model LIM3 applies an ice thickness distribution framework to deal with meter-scale variations in ice thickness (Rousset et al., 2015). Unlike its earlier version (e.g., LIM2), LIM3 allows for five ice thickness categories to account for the non-linear dependence of sea ice processes, in particular growth and melt, on ice thickness.

**Table 1.** List of experiments, reference data sets and variables in forecast skill assessment

| Name | Data sources | Quantities evaluated | Ensemble size |
|------|--------------|----------------------|---------------|
| REF | ERAI for atmosphere | TAS* | 1 |
| | ORAS5 for ocean and sea ice | SIC(SIE), SIT(SIV) | 5 |
| FREE | CMIP6 historical, no anomaly init.(AI) | SIC(SIE), SIT(SIV), TAS | 5[†] |
| AI0 | AI to ocean | SIC(SIE), SIT(SIV), TAS | 5[‡] |
| AI1 | AI to ocean+SIC | SIC(SIE), SIT(SIV), TAS | 5 |
| AI2 | AI to ocean+SIC+(SIT+SNT) | SIC(SIE), SIT(SIV), TAS | 5 |

* TAS is a MIP variable, defined as air temperature at 2 m. [†] FREE are five members from the 25-member ensemble of the CMIP6 historical simulations, initialized from different states (r1, r4, r5, r8 and r18) selected from the 500-year pre-industrial control with EC-Earth3 (piControl, r1i1p1f1 see Fig. S1a). r5 is one of the FREE ensemble, referred to as FREE1; [‡] AI0 (also AI1 and AI2) consists of 5 ensemble members, initialized by 5 sets of ocean and sea ice states, which are based on idential model climatology of FREE1 with anomalies from respective 5 ORAS5 ensemble members. SNT denotes snow thickness.

The EC-Earth3 has been used to perform the CMIP6 historical (1850-2014) and future (2015-2100) scenario simulations with 25 ensemble members following the CMIP6 protocol (Eyring et al., 2016) by the EC-Earth consortium. We arbitrarily select one member (r5i1p1f1, hereafter referred to as FREE1) from the ensemble to obtain the model climatology for the ocean and sea ice used in the anomaly initialization. As there are five members of the initialized hindcast simulations (see Table 1 and sections 2.2), we select additional 4 members besides FREE1 from the 25-member uninitialized simulations to comprise

a 5-member ensemble (hereafter referred to as FREE) in order to have a fair assessment of the forecast skill. The members of FREE are selected with consideration to represent well the overall feature (mean and variability) of the full ensemble (Fig. S1).

The AI method in decadal climate prediction was formulated by Pierce et al. (2004). This approach has already been applied to initialize EC-Earth2.3 decadal predictions contributing to CMIP5 (Hazeleger et al., 2013). A brief comparison of the two generations of EC-Earth decadal prediction system with AI is provided in Table S1. For the previous exercises, ocean-state

anomalies were taken from the product of the ECMWF Ocean Reanalysis System 4 (ORAS4), while sea ice anomalies were obtained from a stand-alone simulation with the ocean-sea ice component of the EC-Earth2.3. For the current study, we derive anomalies from the ECMWF Ocean Reanalysis System 5 (ORAS5).

ORAS5 (See details in Zuo et al., 2019) is a global ocean-sea ice ensemble and its five ensemble members are used to account for observations uncertainties and analysis errors in the surface forcing. In this study period, the ensemble spread for

sea ice is found to be representative for the analysis errors. Compared to ORAS4, ORAS5 increases model resolution from $1°$ to $0.25°$ horizontally and from 42 to 75 levels vertically and assimilates updated observation data sets, such as sea ice satellite data since 1979, so ORAS5 has advantages in providing a physically consistent ocean and sea ice states, and in better representing SST climate state and variability. In comparison with other ocean-sea ice reanalysis (Chevallier et al., 2017), ORAS5 represents Arctic sea ice reasonably well and the errors in SIV (up to 10 %) are comparable to the uncertainties in PIOMASS (Schweiger

et al., 2011). Tietsche et al. (2018) found there is good agreement between SIT from ORAS5 and from L-band observations for thin ice in the freezing season (October–December) with respect to the interannual variability and trends of thin sea-ice area over the pan-Arctic. Therefore, it is a reasonable choice to apply ORAS5-SIT to initialize the decadal prediction, which typically starts on 1 November.

**Table 2.** Definition of initial/forecast- climatology and anomaly

| Categories | Climatology | Anomaly |
|---|---|---|
| Initialization | REF[a]: Nov. 1, averaged over the period 1979-2014, obtained by linear interpolation using the monthly mean of October and November | REF[a]: Nov. 1, daily$^{Y=1979,...,2018}$ - climatology$^{REFa}$ |
| | FREE1: the same as REF[a], but for FREE1 | REF[a] |
| Forecast skill assessment | REF[b]: monthly mean, 20-year mean over the period 1997-2016 | REF[b]: monthly$^{Y=1997,...,2016}$ - climatology$^{REFb}$ |
| | FREE: the same as REF[b], but for FREE | FREE: monthly$^{Y=1997,...,2016}$ - climatology$^{FREE}$ |
| | AIs: the same as REF[b], but for forecast at lead years [1-10][†], respectively | AIs: as FREE, but for respective forecast lead years |

REF[a] and REF[b] are taken from the same REF dataset as in Table 1, but covering different periods.

[†] The first year forecast climatology for the period 1997-2016, denoted as Y1, are calculated using the first-year forecasts from hindcast experiments initiated every year on November 1, for 1996-2015 (marked by the two red triangles in Fig. 3, while the climatology of 10-year lead time (Y10) are calculated using experiments initiated on November 1, for 1987-2006, which means the forecasts were initialized 10 years prior to 1997-2016.

In anomaly initialization, the initial state is generated using the reanalysis state but replacing its climatology with the modelled climatology at a starting date (i.e. climatology of November 1). Here both climatologies are calculated as an average of the climatological monthly means between the two nearest months (i.e, October and November), over the period 1979-2014 for ORAS5 (FREE1), as can be seen in Table 2. Horizontally, ORAS5 data is bilinearly interpolated from 0.25 ° to 1 ° ORCA grid. The initialized variables in the ocean model are three dimensional temperature and salinity. To avoid initial inconsistency in the large-scale dynamics of the system, we do not initialize horizontal velocities. This is a common approach for initialized hindcasts/predictions (see Table 1 in Polkova et al., 2019).

The sea ice variables initialized in EC-Earth3 are ice concentration, ice and snow volume (denoted as $A^{ice}$, $V^{ice}$ and $V^{sn}$) in five thickness categories at a grid-cell level. However, SIC, SIT and snow thickness (SNT) from the ORAS5 reanalysis are single values for the grid-box mean as they are assimilation products using the sea ice model LIM2. Therefore, the volume from ORAS5 is first calculated by multiplying anomaly-corrected ice or snow thickness ($H^{ice,sn}$ in m) with sea ice concentration ($A^{ice}$ in fraction of the grid-cell area). To derive anomaly-corrected $A^{ice}$ (or $H^{ice,sn}$) values we add the anomalies of ORAS5 ice concentrations (or thickness) to the climatology of FREE and then split this corrected field into different thickness categories (see eq.1-5). When anomalies are added to the model climatology, $A^{ice}$ can in some cases violate the valid range [0-1]. Therefore, a few adjustments are made: 1) if $A^{ice} \leq 0$ but $A^{\text{FREE1}} > 0$, then $A^{ice} = 0.05$ and the ice (snow) thickness $H^{ice} = 0.1$ m ($H^{sn} = 0.01$ m); 2) if $A^{ice} \geq 1$, then $A^{ice} = 0.997$; 3) adjust $V^{ice,sn}$ again using $A^{ice}$ and $H^{ice,sn}$ obtained in step (1-2). In this study, sea ice initialization with AI is limited to the region north of 30° N and leaves no modification to the southern area in FREE1.

A major challenge is to distribute the sea ice variable given as one category produced by LIM2 in ORAS5 into five categories in LIM3 in EC-Earth3. Previous studies have explored different solutions in this regard. In sea ice seasonal forecasts, some assimilate satellite SIC in a multivariate data assimilation scheme so as to update SIT instead of directly assimilating it (Massonnet et al., 2015; Kimmritz et al., 2018), while others use forecast tendencies, namely maintaining the distribution of volume between the categories, by multiplying each category volume with the ratio of observed over modelled mean SIT or similarly by nudging towards observations across each category (Allard et al., 2018; Blockley and Peterson, 2018). However, there is no unique solution, since the multiple sub-grid scale configurations can be compatible with one total SIV. For decadal prediction with EC-Earth3-CPSAI, we develop a novel method with 1) a weighting function mapping single-category $A^{ice}$ onto multiple categories; 2) a multi-category thickness distribution depending on concentration levels; 3) both are used when converting the initial volume (i.e. $V^{ice,sn}$) at the grid-cell level to its sub-grid, while thickness in the last category is determined with a constraint of $V^{ice,sn}$ being conservative.

For each grid point with horizontal index (x,y), the initialized $A^{ice}(x,y)$ will be splitted into different categories $g_l^{ice}(x,y)$ as Eq.(1), where $l = 1, ..., L$, denotes the ice category with a total number L=5 in LIM3. In Eq.(2), a weight-likelihood function is derived from the modelled $A^{\text{ctrl}}$ in a 300-year pre-industrial control run with EC-Earth3 (denoted with superscript "ctrl" hereafter) and the data was calculated for November 1 by averaging October and November monthly means. In Eq.(2), $tn$ is the time index over 300 years, at which $A^{\text{ctrl}}(x,y,tn)$ has the minimal difference from $A^{ice}(x,y)$. We assume that based on the same EC-Earth3 model version, $g_l^{ice}(x,y)$ is likely regulated by the weighting function $weight_l^{\text{ctrl}}(x,y)$ given in Eq.(2), and

determined by Eq.(3).

$$A^{ice}(x,y) = \sum_{l=1}^{L} g_l^{ice}(x,y). \tag{1}$$

$$weight_l^{\text{ctrl}}(x,y) = g_l^{\text{ctrl}}(x,y,tn)/A^{\text{ctrl}}(x,y,tn). \tag{2}$$

$$g_l^{ice}(x,y) = weight_l^{\text{ctrl}}(x,y)A^{ice}(x,y). \tag{3}$$

The initialized $V^{ice}$ at the local grid-point (x,y) is calculated as a product of the initialized $A^{ice}(x,y)$ and $H^{ice}(x,y)$ (in fraction of the grid-cell area and m, respectively), which can be splitted into each category $l$ as in Eq.(4). Here $h_l^{ice}$ denotes ice thickness at each category in [m] and $g_l^{ice}(x,y)$ is in fraction [0-1].

$$V^{ice}(x,y) = \sum_{l=1}^{L} g_l^{ice}(x,y)h_l^{ice}. \tag{4}$$

$$v_L^{ice}(x,y) = V^{ice}(x,y) - \sum_{l=1}^{L-1} g_l^{ice}(x,y)h_l^{ice}. \tag{5}$$

We note that in Eq.(4) and (5), $h_l^{ice}$ does not change with geographic location and time, but depends on in which bin $A^{\text{ice}}$ falls (ranging from 0.1 to 1 at intervals of 0.1 in Fig. 1). The relationship between $h_l^{ice}$ and the total ice concentration $A^{\text{ice}}$ is derived from the 300-year control run. We assume the distribution of $h_l^{ice}$ on $A^{\text{ice}}$ identical in the decadal experiments and the control run. Fig. 1 shows that the thickness distributions for $h_l^{\text{ice}}$ ($l = 1, ..., 4$) are quite robust within each bin, and that the more ice-covered (e.g. $A^{\text{ice}} > 0.7$), the lower variance in thickness, in other words, lower probability to melt and shift to neighboring bins. We neglect SIT initialization when $A^{\text{ice}}$ is below 0.1, both because statistically these grid points only account for 8 % of total ice-covered ones and because an observation error of 10 % is often assumed while assimilating SIC (Mathiot et al., 2012).

We aim at imposing local SIV anomalies to LIM3 while keeping the sum of volume over all thickness categories unchanged as in Eq.(4). Therefore, except for the last category, $h_l^{ice}$ ($l \leq 4$) are determined by the corresponding bin of $A^{ice}$ following Fig. 1, while the volume in each category is determined by $v_l^{ice}(x,y) = g_l^{ice}(x,y)h_l^{ice}$. Then the residual of $V^{ice}(x,y)$ will be accommodated in the last category ($L = 5$) as $v_L^{ice}$ in Eq.(5) and $h_L^{ice}$ is contingently resolved. This method imposes the combined anomalous signals of SIC and SIT to sea ice initialization. The same method is applied to discretize snow volume with a multi-category snow thickness on $A^{\text{ice}}$ relationship (not shown).

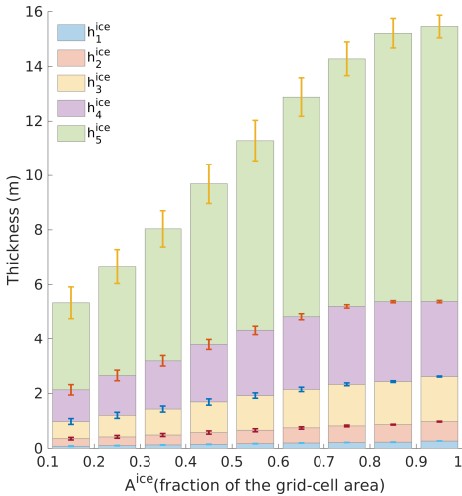

**Figure 1.** Multi-category ice thickness $h_l^{ice}$ ($l$ =1,...,5) distribution on total ice concentration $A^{ice}$ [0.1 - 1] at intervals of 0.1 with mean (column) and standard deviation (error bar), based on the knowledge from a 300-year climate simulation with pre-industrial forcing. It is used to discretize $h_l^{ice}$, if the total ice concentration $A^{ice}$ and volume $V^{ice}$ at a grid cell $(x, y)$ is known from anomaly initialization. For example, for all grid cells $A^{ice}$ falling in the bin 0.1-0.2, $h_l^{ice}$ ($l$ =1,...,5) is 0.08, 0.27, 0.62, 1.15 and 3.20 m, respectively. The ice volume at respective category ($l$ =1,...,4) is determined by $v_l^{ice}(x, y) = g_l^{ice}(x, y)h_l^{ice}$, where $g_l^{ice}(x, y)$ is splitted from $A^{ice}(x, y)$ to each category by using a weight-likelihood function (see Eq.1-3). For the 5th category, $h_5^{ice}$ is determined by $v_5^{ice}(x, y)/g_5^{ice}(x, y)$, where $v_5^{ice}(x, y) = V^{ice}(x, y) - \sum_{l=1}^{4} g_l^{ice}(x, y)h_l^{ice}$ in Eq.(5). Note that the grid-box mean thickness $H^{ice}(x, y) = V^{ice}(x, y)/A^{ice}(x, y) = \sum_{l=1}^{5} v_l^{ice}(x, y)/\sum_{l=1}^{5} g_l^{ice}(x, y) \neq \sum_{l=1}^{5} h_l^{ice}$. Therefore, $H^{ice}$ is regulated by both $V^{ice}$ and $g_l^{ice}$ at the local grid.

A consistency check, also called "sanity check", is carried out for the ICs of the ocean and sea ice model, in order to adjust water and heat flux-relevant variables in the surface boundary fields to the initialized $g_l^{ice}$, $v_l^{ice}$ and $v_l^{sn}$ in a physically consistent way. The method has been used in different sea ice prediction systems (Massonnet et al., 2015; Kimmritz et al., 2018). To complete the initialization, an one-day spin integration is performed with a reduced time-step of 100s and introducing a mask of coastal water (< 100 m deep) to neglect SIT initialization there (indicated in Fig. 2b). This is to ensure dynamical consistency in the initial states and avoid numerical instabilities. The practice of masking some locations out in SIT initialization has been used in many studies, for example by introducing a threshold of SIC>40 % to implement full-field initialization (Blockley and Peterson, 2018) or confining modification of SIT to the Arctic basin with a geographic weighting mask to discard initialization changes in the marginal ice zone (Blanchard-Wrigglesworth et al., 2017). We note that the dotted regions cover a considerable area in the Laptev/East Siberian Seas (Fig. 2b), where SIT observation uncertainties are quite high and are typically excluded from the analysis (Kwok, 2018; Xie et al., 2018). Furthermore, because the regions are highly dynamic and covered by first year ice (FYI, Tilling et al., 2018), the added skill of local initialized SIT is not expected to last over the melting season. In order to identify the added skill from MYI-SIT in the CAO to its adjacent waters on decadal time scales, the ORAS5-SIC

anomaly and FREE1-SIT ICs are implemented to the dotted regions in AI2 as in AI1. In this way, the skill difference between AI2 and AI1 in years 2-9 should arise from a remote (MYI) origin.

## 2.2 Sensitivity experiments with sea ice initialization

The above described AI approach for both ocean and sea ice ICs is hereafter referred to as AI2. A five-member ensemble of the AI2 experiment is performed in this study, with 5 sets of ocean and sea ice ICs generated using the anomalies of five individual members of ORAS5, respectively. Because initial atmospheric conditions do not play an important role for interannual to decadal predictions, for convenience, ERA-Interim (hereafter as ERAI, Dee et al., 2011) is applied as atmosphere ICs for all initialized experiments with FFI. The same initialization strategy with AI to ocean and sea ice and FFI to atmosphere
has already been performed for the EC-Earth2.3 decadal experiments (Hazeleger et al., 2013; Volpi et al., 2017). Initialized ensembles of predictions (re-forecasts) start yearly on November 1 for the period from 1979 to 2018 (a total of 40 start dates) and run 2 months plus 10 years long with the external forcing following the CMIP6 DCPP protocol for dcppA-hindcast and dcppB-forecast experiments (Boer et al., 2016). We generated 10 additional members by means of perturbed atmospheric ICs. This whole ensemble with a total of 15 members states a contribution to CMIP6 DCPP with EC-Earth3-CPSAI (see Table
S1). However, for this study only the first 5 ensemble members (with unperturbed atmospheric ICs) are used and compared to complementary 5-member ensemble predictions described in the following.

Our primary interest is the impact of Arctic sea ice on decadal prediction skill in the last two decades, which is expected to be more representative for the coming decades than an Arctic with large amounts of thick MYI in the last century. In order to investigate the sensitivity of decadal prediction skill to SIT initialization in EC-Earth3-CPSAI, two more initialized ensemble
experiments are performed with ocean-only initialization (AI0) and ocean plus SIC initialization (AI1). We only performed 5 members with AI0 and AI1 for sensitivity analysis, and thus for a fair comparison this study focuses only on 5 members for AI2 and FREE, too. For AI2, our assessment here uses the five members initialized with the five members of OARS5. FREE consists of FREE1 and other four members from the 25 member ensemble of the EC-Earth CMIP6 historical (1979-2014) and the corresponding "medium" Shared Socioeconomic Pathways SSP2-4.5 forcing of ScenarioMIP (2015-2017, Boer et al.,
2016). These four members are deliberately selected to represent the wide range of natural variability in the EC-Earth3 CMIP6 control experiments from which the ensemble of EC-Earth historical simulations starts (Fig. S1a). An assessment of the overall feature of FREE shows no significant difference between FREE and the full ensemble of 25 members (e.g. Fig. S1b), even though the regional differences could be large. A summary of all experiments is given in Table 1. The ensemble-mean of the AI experiments versus that of FREE are used to evaluate the impact of the respective initialization approach. The benefit from
ocean initialization (AI0) is known for the Arctic MIZ (Volpi et al., 2017; Dai et al., 2020), hence we will not address it here. Instead AI0 is used to compare with AI1 to assess the added skill with SIC initialization, and the difference between AI2 and AI1 is used to evaluate the skill gained by SIT initialization.

## 2.3 Skill assessment

As reference fields (REF), the climatological annual and seasonal means are calculated as 20-year averages for the period from
1997 to 2016 based on monthly means of ORAS5 for sea ice and SST and ERAI for TAS (Table 1), respectively. We note that
the reference data (ORAS5 and ERAI) have been produced by assimilation of observations into NEMO and IFS and are thus
not fully independent from EC-Earth3. It is mainly because 1) ORAS5 and FREE are both in ORCA grids, which avoid spatial
errors potentially being either masked or enhanced by remapping from observation- to model-grid; 2) the aggregated quantities
(e.g. SIE and SIV) are not sensitive to models; 3) similar to Volpi et al. (2017), our main focus is to evaluate the relative skills
between different initialization methods with EC-Earth3-CPSAI, compared to the skill of FREE.

Our assessment is based on forecast anomalies (Table 2), rather than absolute errors. The skill assessment always uses the
full 20-year period 1997-2016 for the hindcasts of each lead year, e.g. the assessment for lead year 1 (Y1) includes the first
year re-forecasts initialized from 1996,..., 2015, while the assessment for lead year 10 (Y10) includes the 10th year re-forecasts
started from 1987,..., 2006. The 20-year forecast climatology is calculated for individual lead year, therefore the forecast
anomalies vary depending on the respective forecast lead year (following the recommendation for CMIP6-DCPP, see Boer
et al., 2016). This data selection process guarantees the use of all reforecasts data available for the period of interest and at
the same time a consistent estimation of the model and reference climatologies (García-Serrano and Doblas-Reyes, 2012). The
metrics of anomaly correlation coefficient (ACC) and the Root Mean Square Error (RMSE) with respect to REF are computed
for specific lead time, following the method by Volpi et al. (2017). The confidence interval is calculated with a t-distribution
for the ACC, and with a $\chi^2$ distribution for the RMSE. The assessment of temporal development is performed at two separate
lead times: year 1 and years 2-9. The latter is typically used to assess decadal prediction skill (Goddard et al., 2013).

Our analysis focuses on the sea ice state of SIC and SIT in the Arctic (summarized in Table 1). Our assessment for total SIE
and SIV is applied over the Northern Hemisphere region (NH) with a typical threshold 15 % SIC and 0.15 m SIT, respectively,
to exclude the extensive areas of open water (Schweiger et al., 2011). Additionally, TAS over the Arctic is assessed in order
to identify the local response to anomaly-corrected SST/SIC/SIT at different time scales as well as the impact over land due
to different initialization schemes. A TAS index is computed as field average over the polar cap domain, namely the region
north of the 70° N circle (see Fig. 2). The skill of SST is closely related to that of TAS which is representative for the ocean-
atmosphere heat exchange in the open water. We include SST in supplement (Fig. S4 and S7) to support the regional skill
assessment, when comparing the emerging/degradation of skill in SIC and SIT dependent on initialization scheme. The results
of SNT have not yet been included in this study because there are only very few observations on snow depth over sea ice,
leading to large uncertainties in observations and so as evaluations (Tian-Kunze et al., 2014).

According to the dominating physical regimes (Serreze and Meier, 2019), the regions studied are sorted into six groups
in Fig. 2c (abbreviation explained), representing (1-3) the Arctic Ocean in the CAO (the central Arctic 80°N north, CAA
and the BG region), the Pacific Arctic (CS/ESS/LS), and the Atlantic Arctic (KS and BS); and (4-6) the MIZ in the Atlantic
(HB, BAF and LAB) and in the Pacific (BER) and the transition waters between the ice-covered polar seas and the Atlantic
(GIN). The CAO sector is well confined by the climatological September ice edge (see Fig. 2a and 2b), demonstrating the

dominant influence of thick MYI due to geometric constraints of Arctic coastlines. In the Arctic shelf seas, the extent of ice-free conditions depends on ocean heat transport from the Atlantic (Pacific) in winter (summer) and is strongly modulated by the local wind patterns. By contrast, the MIZ sector adjoins the North Atlantic/Pacific Ocean, climatologically covered by thinner seasonal ice in winter (except for the east coast of Greenland receiving thick ice transported from CAO through Fram Strait).

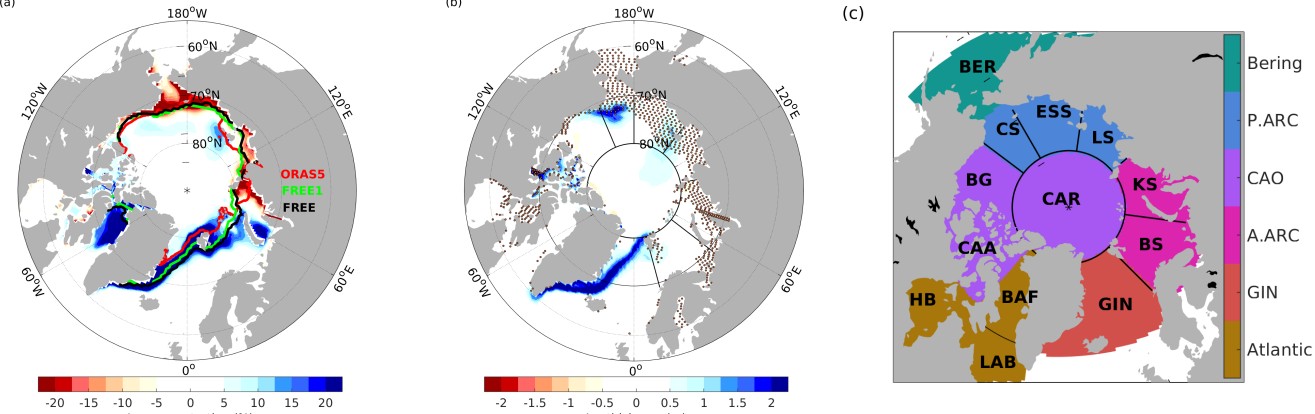

**Figure 2.** Model bias with respect to ORAS5 (REF) on 1 November for the period 1979-2014 for the single-member FREE1. (a) SIC. Color lines indicate the September sea ice extent (15 % SIC) climatology for ORAS5 (red), FREE1 (green) and FREE (black). (b) SIT with dots indicating a mask for area with water depth <100 m where no SIT initialization is applied. The maps have the bounding latitude at 56°N. Blue areas represent regions with more SIC and SIT in the model than REF, reflecting cold bias in surface temperature. (c) Regions considered for Arctic sea ice. Atlantic MIZ (brown): Hudson Bay (HB), Baffin Bay (BAF) and Labrador Sea (LAB); The Greenland/Iceland/Norwegian Seas (GIN, red); Atlantic Arctic (pink): Barents Sea (BS) and Kara Sea (KS); Central Arctic Ocean (CAO, purple): central Arctic (CAR, 80°N north), Canadian Archipelago (CAA), and Beaufort Gyre (BG); Pacific Arctic (blue): Laptev Sea (LS), East Siberian Sea (ESS), and Chukchi Sea (CS); Bering Sea (BER, green).

## 3 Characterization of the initialized climate predictions with EC-Earth3-CPSAI

It is important to measure the time-scale of ICs re-adjustment when using the first seasonal mean for seasonal prediction. In a seasonal prediction system with EC-Earth3, the forecast errors of Arctic sea ice are first attributed to the incompatibilities between the initialized variables, which causes a local dynamical readjustment in a couple of weeks, and then dominated by model inherent bias (Cruz-García et al., 2021). While in a perfect-model seasonal prediction system (no errors in ICs), the forecast errors of SIV can vary with different initialized seasons (Bushuk et al., 2019). Both studies highlight the skill gaps due to two sources of forecast errors. In this section, we characterize the initialized anomaly of sea ice states and the spatial pattern of system errors. We attempt to answer: 1) whether initial forecast errors due to the incompatibility between initialized model variables prevails over one season or a year; 2) after the model drift due to model bias becomes prominent within a decade, would the prediction years 2-9 be representative for decadal prediction in the section 4?

## 3.1 Components of sea ice initialization

The model bias for the initial dates (i.e. November 1) is calculated over the period 1979-2014 between the modelled and reference mean state that are used for obtaining the sea ice anomaly initialization (formulated in Table 2). SIC is generally overestimated (i.e. blue areas dominate over the red ones in Fig. 2a), and particularly there is up to 20 % positive bias in the Atlantic MIZ (BAF,GIN and BS) and 15-20 % negative bias in the Pacific Arctic. In the GIN/BS, the mismatch between the modelled and REF climatology of September sea ice extent indicates the overestimated expansion of MYI from CAO. Along the ice edge of GIN (typically FYI), the positive SIC bias together with 1-2 m thicker ice suggests the role of atmosphere–ocean heat exchange resulting in too fast freezing in autumn, too thick ice in winter and too little ice in summer as inferred from Fig. S2 and S3 for the period 1997-2016. Compared to FREE1, FREE (i.e. ensemble mean) shows similar patterns of model bias but with increased magnitudes (due to a large spread in the FREE ensemble, not shown).

Figure 3 shows differences between anomalies of REF and of FREE1 on the initial date November 1 in each year during 1979-2018 aggregated for the NH SIE and SIV. For REF, the positive anomaly of SIV has reduced by 2/3 from ∼15 to ∼5 thousand km$^3$ in the early 1980s. Since then, the anomalies of SIV and SIE both declined linearly with years and shifted from positive to negative values in the early 2000s, with respect to the mean of the whole period 1979-2014 used for initialization. The 20-year averages of anomalies from 1996 to 2015 are -0.11 million km$^2$ for SIE and -3 thousand km$^3$ for SIV, respectively. In comparison of the 20-year averages between REF and FREE1, there is a relative large difference in SIE (-0.11 versus -0.77) but little difference in SIV (-3 versus -3.7), suggesting the retreat of SIE more rapid than the thinning of SIT in FREE1 in response to Arctic warming. In other words, the modelled sea ice is characterized by freezing too fast in autumn, too thick ice in winter and too little ice in summer as inferred from Fig. S2.

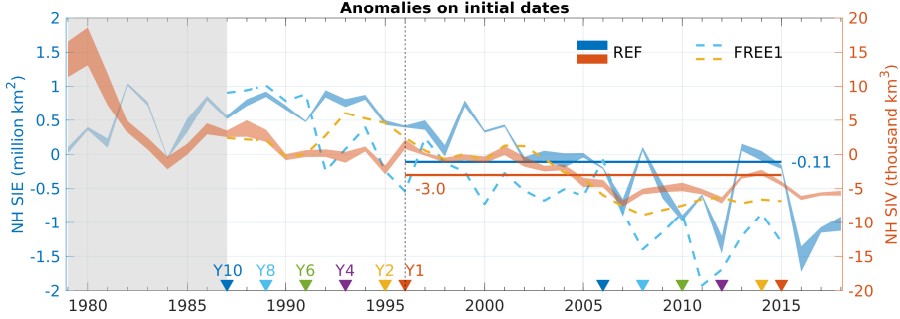

**Figure 3.** Anomalies of SIC (SIT) on 1 November aggregated for the NH sea ice extent (volume) in blue (red) during 1979-2018. The ensemble spread (between min and max) of REF is presented by filled area, while anomalies of FREE1 are presented by dashed lines. For AI2, the 20-year averages of anomalies taken from REF for lead year 1 (Y1, between two red triangles) are -0.11 million km$^2$ for SIE and -3.0 thousand km$^3$ for SIV (red and blue straight lines), respectively. Note that the skill assessment period is from 1997 to 2016 (see Table 2), where Y1 denotes forecast initialized from 1996, ..., 2015 and Y10 denotes the 10th year forecast initialized from 1987, ..., 2006. Thus the anomalies are more positive for predictions with longer lead years assessed, as can be seen when compared to those in Y1.

Figure 3 depicts different sea ice initialization strategies, e.g. in the first year low-SIC (FREE1) and high-SIT (FREE1) for AI0 in contrast to high-SIC (REF) and low-SIT (REF) for AI2. AI1 has the combination of high-SIC (REF) and high-SIT (FREE1). We note that AI0 and AI1 are initialized with different combinations of ocean (REF) and sea ice anomalies (REF or FREE1) while AI2 is not. As model physics differ, initialization shocks likely impact the prediction skill. In the meantime system errors (not corrected in anomaly initialization) may develop fast, if the REF anomalies of FYI (<10 % SIC and 0.2 m

SIT) are much smaller than the positive (negative) bias (>20 % SIC and 1 m SIT), such as in the Atlantic (Pacific) sector in Fig. S2. Therefore, it is essential to track the development of initialization shocks and system errors in the next section.

## 3.2    Forecast drift

Figure 4 shows the monthly forecast error of SIE, SIV and TAS for different initialization methods as a function of forecast lead time in the 10-year prediction. The monthly forecast error is determined by subtracting the climatological monthly mean

of REF for the period of January 1997 to December 2016, rather than forecast anomaly as defined in Table 2. To compare, the monthly biases of FREE (FREE1) for the respective variables were shown as gray (black dashed) lines and filled areas in Fig. 4a-c, which are simply repeated annual cycles averaging for the same period (1997-2016) with annual mean bias of 0.4 (-0.2) million $km^2$, 5.6 (2.3) thousand $km^3$ and -0.7 (0.1) K for SIE, SIV and the Arctic cap TAS, respectively. It is by chance that FREE1 (black dashed line) has smaller bias in SIE, SIV and TAS over the Arctic than the ensemble mean of FREE (thick

gray line in Fig. 4, also see FREE bias in annual min/max sea ice and mean TAS in Fig. S2 and S3).

     A general feature of all three variables is that biases in all initialized experiments (AIs) vary in the range between FREE1 and the ensemble mean of FREE, so that initialization results in a positive annual mean bias to the SIE and SIV (i.e. larger SIE and SIV) and negative annual mean bias to TAS with respect to FREE1, taking the annual mean bias of AI2 (blue dots) as an example. There is a slight tendency that the forecast error increases for longer lead time. The larger biases in AIs experiments

than FREE1 result from anomaly initialization where the model bias of FREE1 is not removed from the initial state, but surplus with more positive anomalies of sea ice for prediction with longer lead years assessed, compared to those in the first year forecast over the period 1997-2016 (i.e. Y1 in Fig. 3). The differences between AIs are generally small after Y1, indicating the forecasts are drifted toward the model climate (as represented by the ensemble mean). And the forecast error for SIE is relatively less sensitive to different initialization than that of SIV and TAS. On average the long-term forecast drift is small as

indicated by the annual mean errors in AI2 (blue dots in Fig. 4). There is a slight decline in both SIE and SIV from Y1 to Y3 followed by a return between Y5-Y7, and a tendency of larger SIE and SIV for longer lead time. Correspondingly, TAS in the initialized hindcasts is gradually pulled to the colder climate over the Arctic cap domain, showing a tendency of negative bias in all AIs in both winter and summer for longer lead time, i.e. Y8-Y10.

     The model biases show rather strong seasonal cycles, with generally smaller biases in winter but much larger in summer

for all three variables in AIs than FREE. The roles of initialization shocks are not evident in all AIs since the first prediction year (from January 1). It is consistent with the results of Cruz-García et al. (2021) that the readjustment between surface ICs takes place within the first few weeks. These results emphasize the importance of drift-correction via correcting the lead-time dependent bias for multi-year prediction skill assessment. Among all AIs, AI1 (pink line) shows the least magnitude of

positive seasonal bias of SIE and SIV from Y6 afterwards, and correspondingly the warmest TAS. It seems that, for longer lead times (>Y5), AI1 performs closest to FREE1. It suggests that AI1 with REF (ocean plus SIC) anomaly but SIT from FREE1 initialization imposes less strong constraints on the development of the sea ice forecast than ocean-only (AI0) and all (AI2) initialization at decadal time-scales.

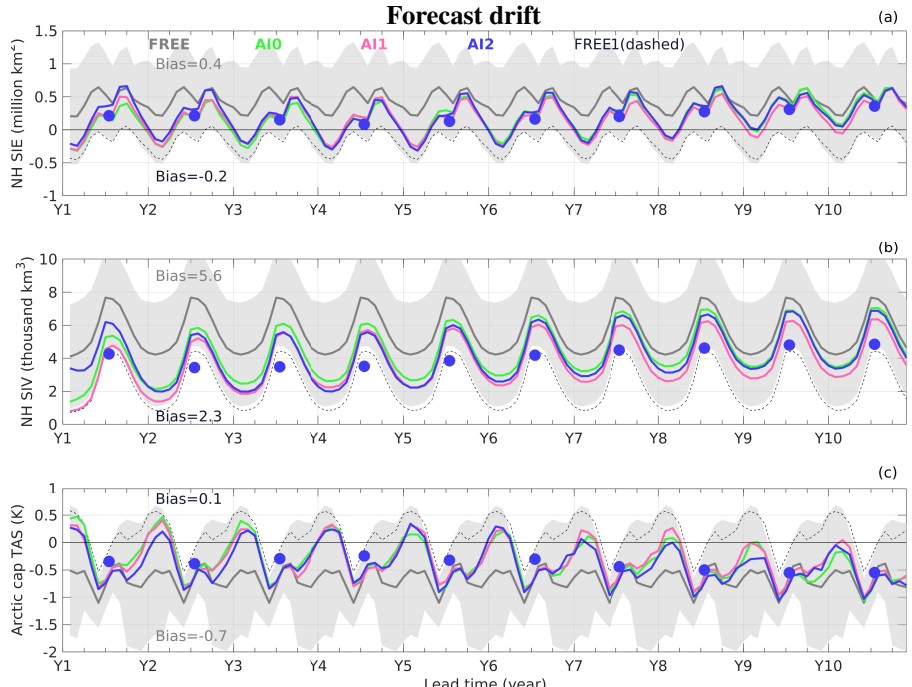

**Figure 4.** The development of forecast drift for hindcast experiments using different initialization methods for the NH sea ice extent (a), volume (b) and TAS over the polar cap domain (c), respectively. The drift is computed as difference of monthly climatology over 1997-2016 between forecast for a specific lead time and REF. The time series is shown as 3-month moving average and therefore the first three months after initialization (i.e. Nov, Dec and Y1-Jan) are masked out. The colored lines plot the ensemble means of the experiments initialized with ocean-only (i.e, AI0, Green), ocean plus SIC (AI1, pink) and ocean plus SIC and SIT (AI2, blue), respectively. The un-initialized experiments FREE are shown as repeating annual cycles for ensemble mean (gray line) and spread ($\pm 1\sigma$, gray shading), as well as the member used for initialization (FREE1 in black dashed line). The annual mean of AI2 are marked with solid dots for each forecast years.

### 3.3 Imprint of initial conditions in the first year

#### 3.3.1 First winter forecast

In this section we examine the immediate benefit (or degradation) from initial anomalies (ICs inconsistency) due to different initialization strategies on spatial scales. The winter mean time series are considered as December-January-February of 1996-

97 to 2015-16 for REF and the forecast of the first winter. When we calculate the anomaly of regional averages from REF (i.e. ORAS5), the Arctic Ocean is fully covered by sea ice during winter and there are neither trends nor interannual variability in SIE (SIC changes below 5 %) and year-to-year changes in MYI-SIT up to 1 m. One exception is the Barents Sea with high variability in SIC because it is open to the Atlantic inflow. In the Atlantic/Pacific MIZ, the warm Atlantic/Pacific waters regulate changes in the very thin FYI states (below 10 % SIC and 0.2 m SIT). As mentioned before, the objective of assessing TAS changes is to identify local changes in ocean-atmosphere heat exchange altered by sea ice initialization.

Figure 5 depicts the ACC for the first winter predictions. The stippled areas indicate where the Pearson correlations are not significant (p=0.05). In the Arctic Ocean, the correlation for SIC (Fig. 5, left) in all experiments (FREE/AI0/AI1/AI2) is low because the area is constantly ice-covered. The correlation for SIT (Fig. 5, centre) is significant high in FREE in the CAO, which is further enhanced from AI0 to AI2, resulting in the areas with significant high correlations expanded in the CAO towards the East Greenland Current (EGC) in AI2. The highest correlation of SIT (AI2) in the CAR/BG reflects the spatial scale with thinning trends of CAO-MYI (Serreze and Meier, 2019), while the skill of EGC in AI2 mirrors the impact of strong winter sea ice outflow from the CAO-MYI, but confined by the large model bias (SIT, Fig. S2b). TAS in FREE shows the largest area with significant positive correlation over the CAO than these with AIs. It indicates that its prediction skill is mostly controlled by external forcing of anthropogenic green house gases (see in Fig. S1b) compared to internal variability (corrected via initialization) during the last two decades.

In the MIZ (Fig. 5), FREE generally shows no significant skill (i.e. low correlation) in SIC and SIT, except some parts of GIN/BS/KS. Coincidentally over the whole regions of GIN/BS/KS, FREE shows high skill in TAS, reflecting the influence of the increasing Atlantic heat inflow since 1990s (Serreze and Meier, 2019). AI0 increases ACC (SIC and TAS) significantly in the BS/HB, while those skill of AI0 in the two regions seems diminishing in AI1 by SIC initializaiton (explained later with RMSE in Fig. 6). Comparing AI2 to AI1, the SIT initialization significantly enhances the high correlation areas for SIC, SIT and TAS in some parts of the Baffin Bay and KS. The major benefits of AI2 (TAS) are seen outside of the polar cap domain, manifested as significantly enhanced correlations over the North Atlantic Ocean.

Figure 6 illustrates the RMSE of FREE with respect to REF, and the RMSE skill scores for SIC, SIT and TAS. The largest RMSEs of up to 25 % SIC and 1 m SIT can be seen in the MIZ (Figure 6, top row), mirroring the system errors (i.e. bias) developed between autumn (1 November, Fig. 2) and late winter (March, Fig. S2). The largest RMSE is due to overestimated anomalies for SIC in the Atlantic/Pacific MIZ and for SIT in the Arctic shelf seas coasts and the GIN Seas. The largest RMSE in TAS over the Arctic Ocean is consistent with those of SIC. In order to identify the benefit of specific initialized model components, we evaluate the RMSE skill score (RMSESS), which compares the RMSE of AI2 ensemble to the other experiments, i.e.

$$RMSESS = 1 - RMSE_{AI2}/RMSE_{INIT} \tag{6}$$

where *INIT* denotes the different experiment to be compared, i.e. FREE, AI0 or AI1, respectively. A positive RMSESS indicates better accuracy (smaller RMSE) of AI2 compared to *INIT*. Specifically, a positive RMSESS relative to FREE (AI2/FREE hereafter) indicates benefit from both ocean and sea ice initialization, the RMSESS relative to AI0 (AI2/AI0 hereafter) indicates

benefit from sea ice (SIC and SIT) initialization, while the RMSESS relative to AI1 (AI2/AI1 hereafter) singles out the benefit of SIT initialization.

In the CAO, the skill score for SIC (∼0.05 in Fig. 6, left column) appears to be unaffected by any initialization, with respect to RMSE nearly 0 in FREE. For SIT (Fig. 6, centre, lower three), RMSESS>0.2 is commonly found in the CAO and its adjacent waters (GIN/BS/KS). Combining the gradually enhanced correlation from AI0 to AI2 (SIT) in Fig. 5, the gained skills can be linked to the initialized MYI-SIT, highly correlated with external forcing. For the same reason, the correlation in Chukchi Sea is found enhanced from AI0 to AI2 in Fig. 5, which results in high scores in AI2/AI1 (SIT). Note that the stippled areas (e.g. ESS/LS/KS) indicate no difference in sea ice ICs between AI1 and AI2 (reason in Section 2.1). By linking the skill changes in the local TAS, the degradation of SIT in the ESS/LS is probably attributed to advection of corrected SIT from the Chukchi Sea driven by local wind pattern, which prevails over external forcing, while the improved skill (SIT) in KS may originate from its neighboring waters (CAO/BS) with corrected SIT. FREE is best for TAS in large parts of the CAO with respect to both ACC and RMSE, presumable because the atmospheric large scale circulation in all initialized experiments is undertaking adjustment to the initialized states in the first few months.

In the MIZ (Fig. 6), AI2/FREE (SIC) is generally positive (benefit from all init.), except negative (less skillful) in the Bering Strait and EGC, which coincides with the maximum RMSE (SIC) in FREE. Skills (SIT) are degraded along the ice edge of the Labrador/GIN Seas in AI2/FREE (also in AI1/FREE and AI0/FREE, not shown), showing opposite skill changes with positive score (>0.5) in both AI2/AI0 and AI2/AI1 versus negative score (<-1) in AI2/FREE. The negative scores in AI2/FREE (SIT) coincide with the maximum bias of sea ice states (Fig. S2a and b), suggesting the major role of model bias in causing forecast errors. On the contrary, the model bias (Fig. S2 and S3) is negligible in the Hudson Bay and the improvements (SIC and SIT) are only seen in AI2/FREE (RMSESS>0) but not in AI2/AI0 and AI2/AI1, indicating a dominant role of ocean temperature in shaping the growth of sea ice in this region. This supports the results from Dai et al. (2020) that good skills in winter prediction can be gained by assimilating SST alone in the N. Atlantic FYI regions. The impact of initialization on TAS is mostly outside of the polar cap domain with RMSESS >0.2 in the North Atlantic Sector, Greenland and the Alaska Peninsula, whereas RMSESS<-0.2 in the Pacific Sector and eastward to the Siberian region. Similarities between all RMSESS (TAS) figures suggest that the TAS skills are mostly attributable to the SIT initialization. Differences between AI2/FREE and AI2/AI1 can be inferred that SIT initialization is beneficial to the expansion of thick ice from the CAO to the KS/BS, thus counteracting the Atlantic inflow to the southern BS.

In summary, the thick ice in the CAO during winter shows large variability in SIT (up to 1 m) but almost no changes in SIC (<5 %) . This makes the SIT initialization perform the best in increasing the skill of SIT in a basin scale. In the MIZ, the very thin FYI variability depends on the Atlantic/Pacific heat inflows. There is a direct improvement in RMSESS (SIC) by ocean initialization (AI0) in the Atlantic MIZ, where the model has too much ice. But AI0 works less well in the Pacific MIZ where the model has warm bias. The added value by SIC initialization (i.e. AI1) is found limited in the Atlantic/Pacific MIZ because the anomaly (REF) of sea ice states is much smaller than the system errors. Alternatively, assimilating SST is recommended to effectively constrain the development of model bias, so as to improve the sea ice prediction skill (Dai et al., 2020).

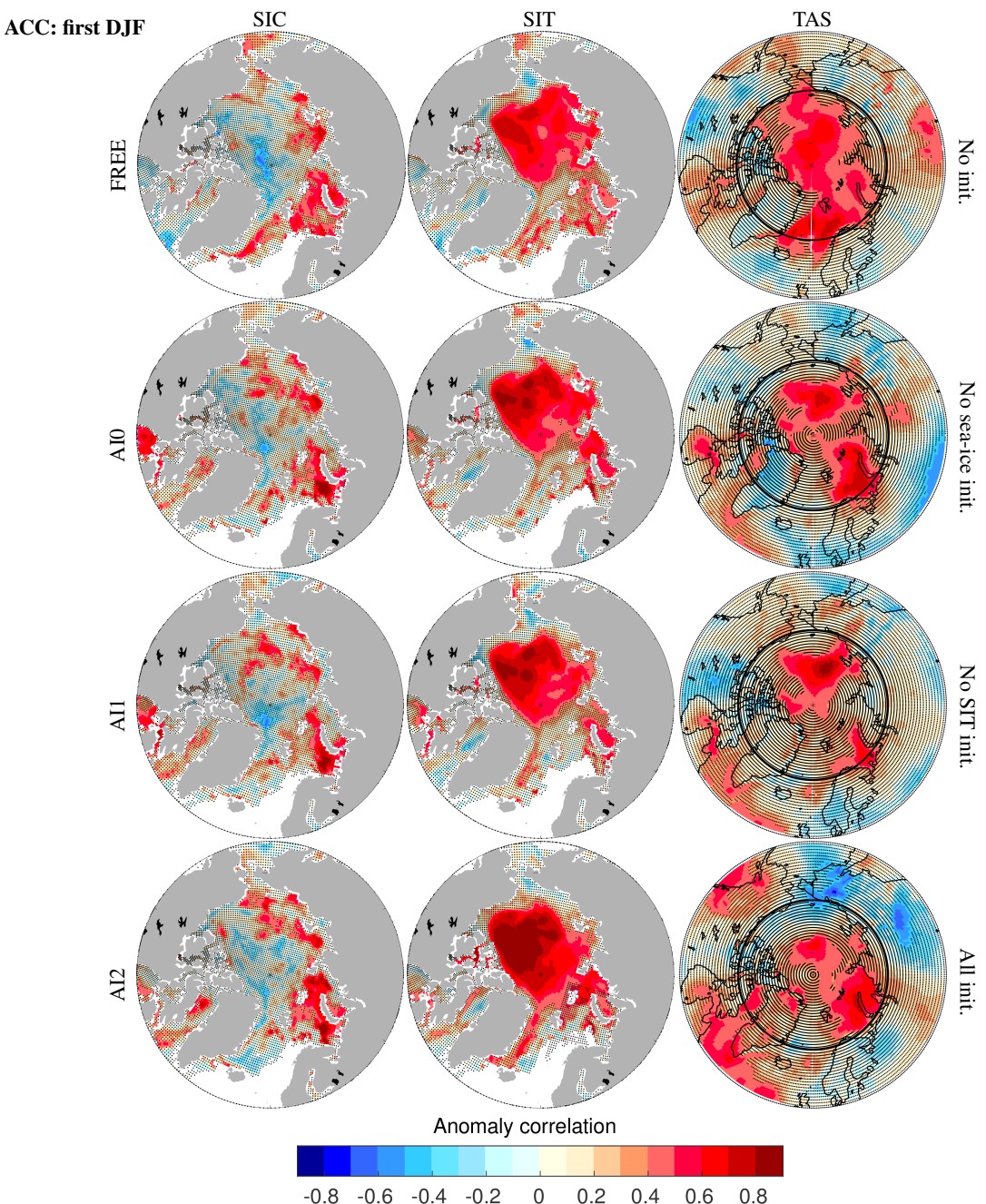

**Figure 5.** Anomaly correlation of the first winter SIC (left), SIT (centre) and TAS (right), respectively for FREE, AI0, AI1 and AI2 experiments from top to bottom. The first winter forecast evaluated here is the DJF-mean after initialization on 1 November, on each year of 1996-2015. The reference data are taken from ORAS5 or ERAI. Regions are stippled if not significant (p = 0.05). The black line illustrates the polar cap domain.

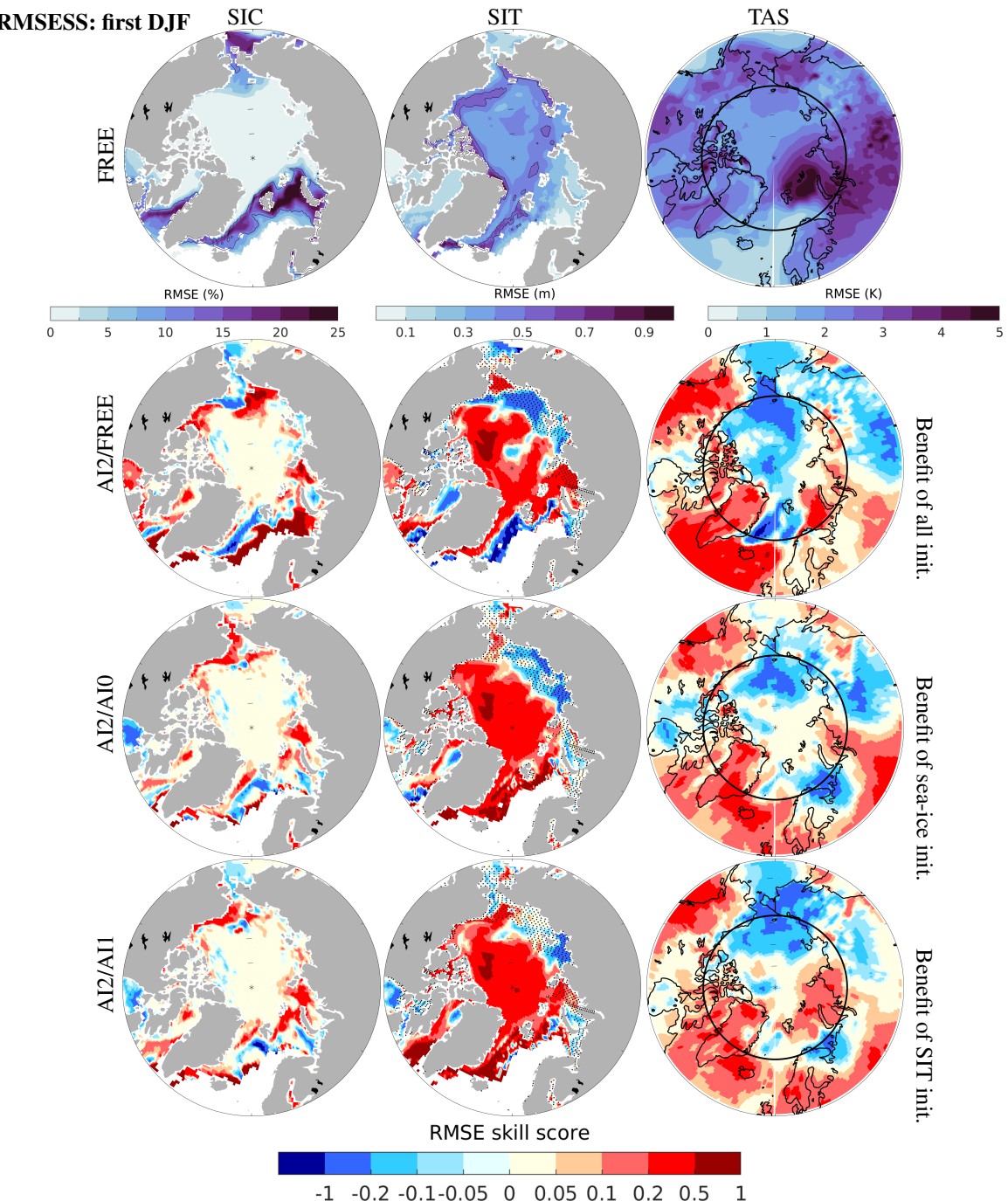

**Figure 6.** RMSE of FREE with respect to REF (top row) and RMSE skill score (lower rows) for the first wintertime (DJF) forecast: SIC (left), SIT (centre) and TAS (right). The contour lines mark RMSE≥10 % or 0.5 m for SIC and SIT, respectively. The RMSE skill score is calculated as 1-(RMSE$_{AI2}$/RMSE$_{INIT}$), where *INIT* denotes FREE (no init.), AI0 (no sea-ice init.) and AI1 (no SIT init.). Scores above 0 denote more accurate in AI2 than *INIT*, and vice versa. Note that the labels of the colorbar for skill score is asymmetric because the minimum of SS can be far below -1 in contrast to the maximum of 1. The stippled areas in the middle columns are the same as in Fig. 2b.

### 3.3.2 The first 12 months forecast

As the persistence timescales of upper-ocean heat content and SIT can be longer than one season, we continue to identify the relative contribution from ocean and sea ice initial constraints as the dominant source of predictability in the first forecast year. The prediction skill in the first 12 months, counted from the start month (i.e. November), is evaluated by analyzing the temporal development of ACC and RMSE relative to REF (as Eq.1-2 in Volpi et al., 2017). The definition of forecast and reference anomalies is given in Table 2. The monthly mean of total SIE and SIV are calculated and extracted from the first 12 months forecast initialized on November 1 from 1996-2015 as indicated in Fig. 3. In the same way, the first 12 months forecasts of TAS is averaged over the polar cap domain (see Fig. 2a) in order to investigate the direct impact from MYI in the CAO on the atmosphere, as most of the region north of 70 °N is sea ice covered year-around in the climatology (Fig. S2c and d). When the correlations of AIs meet, or even fall below FREE in Fig. 7, it means that there is no skill from the initialization any more. On the contrary, when the RMSE of AIs exceeds the FREE one, it indicates no benefit from initialization afterwards. Additionally, the thin lines in Fig. 7 represent the upper/lower bounds of the 95 % confidence intervals obtained with a t-distribution for correlations and a $\chi^2$ distribution for RMSE. The correlation with one experiment is not significant, if the confidence interval goes below 0. Furthermore, the difference between two experiments is not significant, if those two intervals overlap. The results should be interpreted with cautions due to a small sample size (N=20).

The correlation of SIE (Fig. 7a) shows that AI0 and AI1 improve their skills over FREE only in the first month, in contrast to AI2 for a little longer lead time up to 3 months. All confidence intervals overlap, suggesting none of the differences are statistically significant. The similarity in correlation are found between AI0 and AI1 until March and between AI0 and AI2 from April, which suggests a major contribution of SST improvements. As all initialized experiments show a recovery of skill in June-July, the predictive capability for summer SIE possibly comes from pre-winter SST anomalies that could have been stored in the ocean heat content (Holland et al., 2011; Chevallier and Salas-Mélia, 2012). The reduction in RMSE with initialisation is significant until lead month 10 (i.e. August in Fig. 7d). Similar results are also found in the study of Dai et al. (2020) in which the pan-Arctic SIE can be predictable up to 12 months with the NorCPM climate prediction system by only assimilating SST.

Compared to SIE, the skill in SIV (Fig. 7b) has been improved significantly by all initialized experiments in the first five months (i.e. March). In the first two months, AI2 clearly outperforms other experiments due to SIT initialization. Afterwards, there is no significant difference between initialized experiments, indicating the major contribution of SST improvement. Since April, the skills of all three initialized experiments meet the skill of FREE and degrade slowly with forecast months. Until lead months 12, all correlations are above 0.8, indicating the dominant role of external forcing. In Fig. 7e only AI2 shows the lowest RMSE in all experiments throughout the year. By contrast, there is no significant difference in RMSE between AI0 and AI1, because two experiments apply identical SIT but different SIC initial states. This suggests the relative importance of SIT initialization in constraining SIV anomaly in EC-Earth3.

For TAS, analogous results with SIE are found in Fig. 7 (c, f) that all 95 % confidence intervals overlap, indicating no significant difference between experiments. The improvement by initialization is only significant in the first month. The degradation

in AI0/AI1/AI2 is reflected by Fig. 5 (TAS, right) that the large-scale atmospheric circulations are undertaking changes to different ocean/sea ice surface states in the first winter months. Interestingly, the confidence intervals of all experiments mostly go negative between February and August (marked by open circles), indicating insignificant skill during the melting season, especially AI0. It can be linked to the overestimated thick ice cover by September in the Arctic shelf seas (Fig. S2c and d) that prevent rapid heat exchange between atmosphere and the upper ocean during the melting season. Therefore, it is challenging to constrain the atmospheric states in the first 12 months forecast with sea ice anomaly initialization.

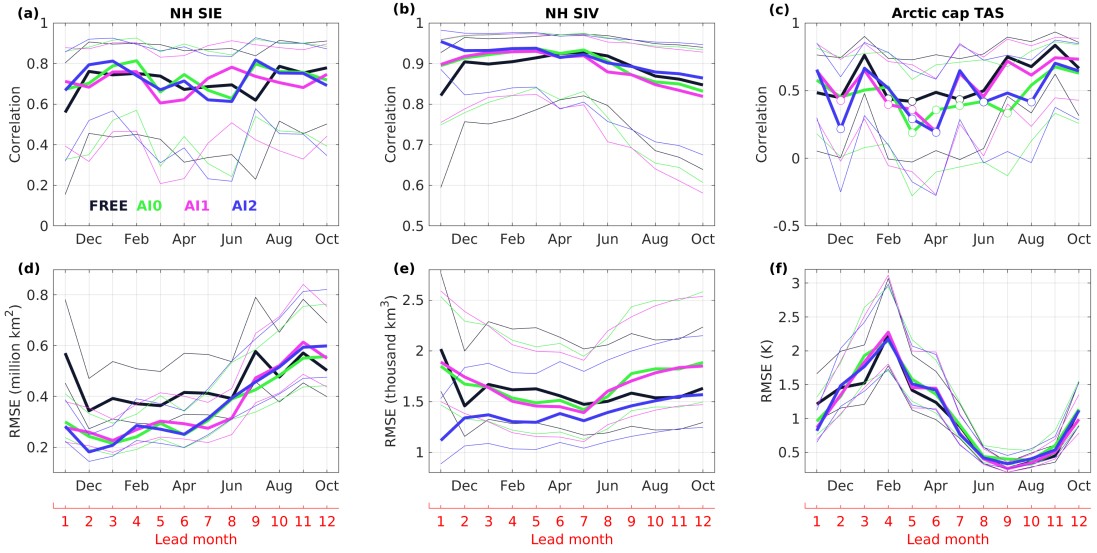

**Figure 7.** Correlation (a-c) and root mean square error (RMSE, d-f) for the NH SIE (million km$^2$), SIV (thousand km$^3$) and TAS (K) over the polar cap domain, against REF for the first 12 months forecasts started on November 1 from 1996-2015. It is calculated for each lead time with respect to its monthly climatology. The thin lines represent the 95 % confidence intervals obtained with a t-distribution for the correlation and a $\chi^2$ distribution for the RMSE. Open circles are used if not significant (p = 0.05). Note y-axis has different scales.

## 4    Decadal-scale skill assessment

### 4.1    Year 2-9 average predictions

In this section we focus on understanding the origin of decadal predictability and the relative contribution from the ocean, SIC and SIT initialization on the decadal time-scale. The initial year is excluded in this section because the imprint of initial conditions is strong and discussed above. Following the verification framework for interannual-to-decadal predictions (Goddard et al., 2013), prediction skill (AI0,AI1,AI2) is averaged over forecast years 2-9 and compared to the respective FREE projection. In contrast to the first winter mean, the temporal smoothing over an 8-year window will typically reduce high frequency noise and increase signals resulting from external forcings such as increase of greenhouse gas concentrations. There are some

evidences of negative trends in Arctic SIE and SIV (Fig. 3) and positive trends in global TAS (Fig. S1) during the recent two decades in the FREE ensemble.

Figure 8 shows that the areas with significantly positive correlation (>0.5) are larger in the year 2-9 than that for the first winter (Fig. 5) for all three variables and all experiments. The high correlation indicates that external forcing with warming trends determines the decadal prediction skill. Therefore, FREE (Fig. 8, top) shows highest correlation in SIC (outskirt of the central Arctic), SIT (BG) and TAS (CAO), likely associated with the trends in local SIV under Arctic warming. In general, AI0 presents a dominant role of ocean initialization in all experiments. The added skill (correlation) by AI1 to AI0 is largest for SIC

in LS and for SIT in ESS, while AI2 improves SIC in the BG. Although SIT initialization was not locally implemented to the ESS, sea ice is moving driven by advective processes or winds (Guemas et al., 2016), which can result in enhanced correlations of SIC and SIT in both AI1 and AI2 at longer lead time. The improvement in TAS follow those of improved sea ice state in FYI regions and expand over land as well.

With respect to RMSE in FREE (Fig. 9, first row), the spatial patterns of SIT and TAS derived from the year 2-9 average are

very similar to those of first winter means shown in Fig. 6 but the magnitude is smaller. For SIC, the errors gradually increase from 5 % at the outskirt of the central Arctic (inside fully ice-covered) to 10 % towards the MIZ with maximum of 15 % in the northern BS, reflecting the effect of the maximum winter and summer bias on prediction skill for SIC (Fig. S2a and c).

Both observations and climate models have suggested that in the MIZ the variability of sea ice is largely influenced by the oceanic heat flux convergence, therefore, the prediction errors can be greatly reduced by advection of improved ocean

temperature whereas little benefit from SIT initialization is expected (Koenigk and Mikolajewicz, 2009; Årthun et al., 2017; Onarheim et al., 2018; Bushuk et al., 2019; Dai et al., 2020). As we discussed in section 3.3.1, the model bias plays a dominant role over initialized anomalies in forecast errors for the MIZ. The relative skills (AI2/FREE, AI2/AI0 and AI2/AI1) for SIC and SIT in year 2-9 (Fig. 9, left two, lower three) is very similar to those in the first winter mean (Fig. 6, left two, lower three).

Compared with the first-winter forecast, the striking skill changes of SIC and SIT in the inner Arctic (Fig. 6 versus Fig. 9,

left, center two) suggests that the imprint of local sea ice ICs are removed on longer lead times. Comparing the center and lower panels of Fig. 9 (SIC, left), the improved skill in AI2/FREE for SIC (RMSESS >0.2, red) suggests that ocean initialization is the most important source of predictive skill at decadal time-scale, whereas the degradation of AI2/AI0 (RMSESS <-0.2 in blue, versus AI2/AI1~0) in the CAR is attributed to corrected SIC. Changes in SIT in the Arctic basin are regulated by ocean circulation such as the BG and the transpolar drift (Davis et al., 2014). The improved skill in AI2/FREE (SIT in Fig. 9, center)

along the Arctic shelf seas and the pathway of transpolar drift suggests the positive effect of increasing Atlantic/Pacific heat inflows and enhanced transpolar drift (Carmack et al., 2015). By contrast, there are negative effects of ocean initialization with degraded skills for SIC and SIT in the BG region with negative RMSESS in AI2/FREE (also AI0/FREE, not shown) and slightly negative in AI2/AI1 in Fig. 9 (lower rows, left two). This may be associated with an increasing polarward expansion of FYI zone in the southern BG (Bliss et al., 2019), where a thinner sea ice cover (represented by local SIT anomaly) will be

more easily forced by wind, and consequently lead to stronger circulation in the BG (Armitage et al., 2020). However, there are substantial system bias in sea ice states in the BG in September (>20 % SIC and 2 m SIT, Fig. S2c and d), therefore, the immediate benefit from local SIT initialization (Fig. 6, centre) cannot hold on time-scales up to a decade in this respect.

**ACC: years 2-9**

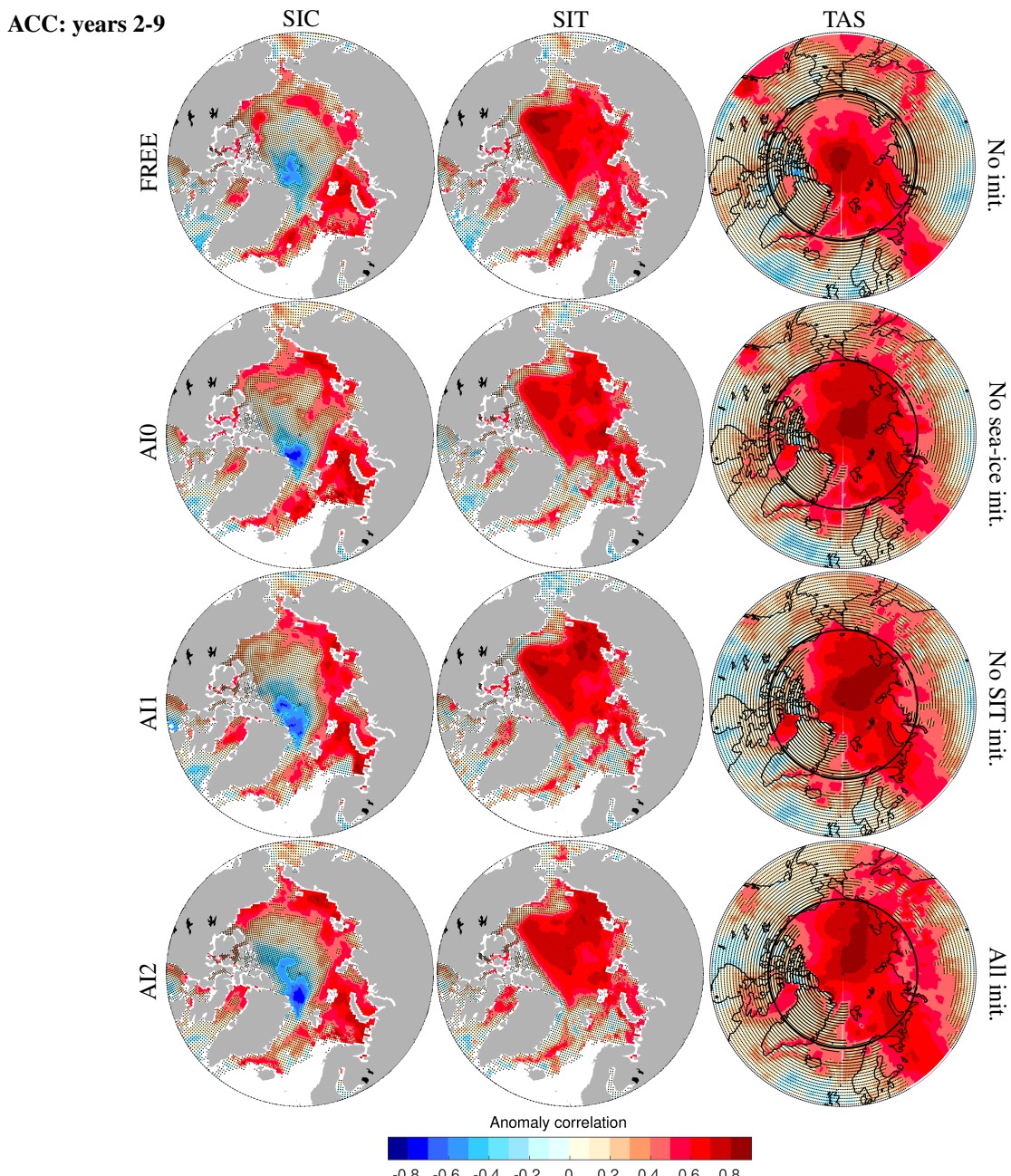

**Figure 8.** Same as Fig. 5, but for years 2-9.

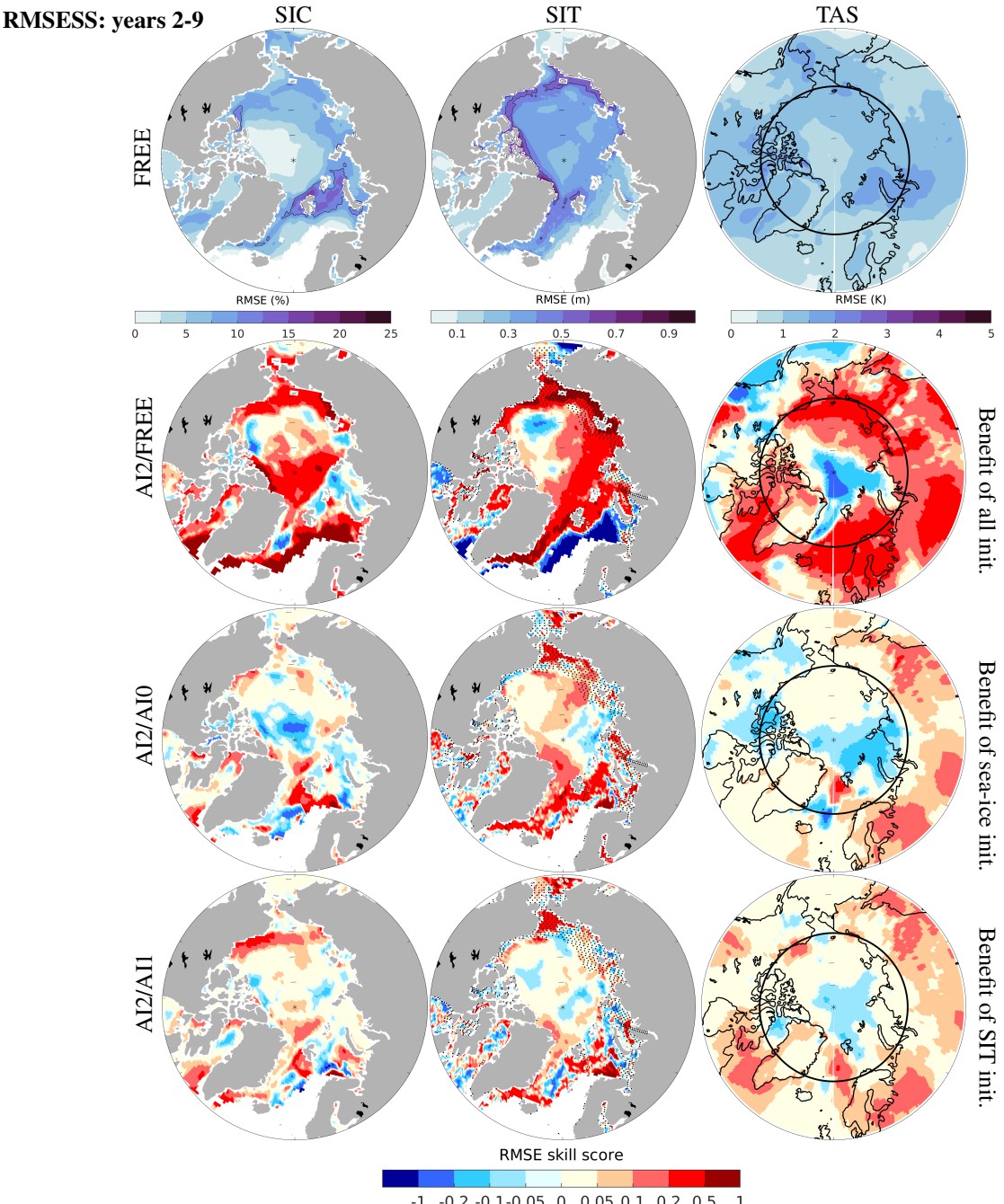

**Figure 9.** Same as Fig. 6, but for years 2-9.

For TAS the area with improved skill in AI2/FREE (Fig. 9, right) is considerable related to increasing ocean temperature of the Atlantic/Pacific inflows (Serreze and Meier, 2019), covering the N. Atlantic, the eastern Arctic and expanded over land. 495 The added skill in AI2/AI1 in the landward vicinity of the BS/ESS may result from changes in local wind pattern due to

the thinning of SIT in AI2 relative to AI1. Interestingly, TAS over the KS is degraded in AI2/FREE and AI2/AI0 (but not in AI2/AI1), indicating a negative effect from both ocean temperature and SIC initialization (AI0 and AI1). We should note that there are remarkable warm biases of the annual mean in SST and TAS for the BS/KS (see FREE1 in Fig. S3a and c), accompanied with a large retreat of summer sea ice in FREE1. It suggests the warm bias of SIC in summer prevails over the cold bias of SIC in winter (Fig. S2a and c) in years 2-9 average prediction. Although we have found the dominant role of ocean initialization in improving SIC and TAS prediction over the BS/KS in winter (Fig. 6), it seems to have an opposite effect in summer. Dai et al. (2020) have shown that the seasonal prediction skill for the September SIE in the BS cannot be gained by assimilating SST (or correcting SIC) alone due to lack of constraint on surface winds. Consistent with their results, SIT initialization seems promising to constrain the summer retreat of MYI (i.e. thick ice) in the northern BS/KS, thus advancing or blocking the Atlantic heat inflow to the KS. By contrast, the cold bias of winter SIC prevailing over that of summer SIC in the Labrador Sea (Fig. S2 versus Fig. S3), so that corrected (thinner) SIT will make the excessive sea ice cover (biased SIC) easier to be driven by winds. In turn, the skill in TAS (via ocean and atmosphere heat exchange) will be improved.

## 4.2   Regional-mean skill for interannual-to-decadal time scales

The results from sections 3.3 and 4.1 provide important insights into the regional variability of sea ice prediction skill at different time-scales. It is clear that the benefit from SIT initialization in shaping the local sea ice development does not last beyond the first summer. However, the added skill of SIT on decadal time-scales in some FYI regions seems associated with remote regions (CAO-MYI) which may last for several forecast years with the support of accurate ocean conditions. In the final analysis, we aim at providing details on the temporal evolution of the skills of initialized hindcasts for the Atlantic/Pacific Arctic shelf seas adjacent to the CAO, with evidence of the expansion of polarward retreating sea ice (Bliss et al., 2019). As the climate warming continues, the CAO might become seasonally ice free in the future. Our goal is to shed light on how the key mechanism governs the decadal predictability of the Arctic sea ice in the coming decades. Therefore, the Barents, GIN, E. Siberian Seas and Beaufort Gyre are selected to represent respective physically-dominated regimes. The FYI regions (HB/BAF/LAB/BER) are of little interest here, as they are heavily influenced by Atlantic/Pacific Ocean dynamics (also system errors) and the role of CAO-MYI diminishes. The regional assessments with groups (Fig. 2c) are shown in Fig. S5-S7. The RMSESS are calculated on monthly basis for each grid points in years 1-9 and then the area means are considered as its regional skill. Regional skill scores of AI0/FREE, AI1/FREE and AI2/FREE are assessed for SIC and SIT with red (blue) colors for positive (negative) value of scores, indicating better (worse) accuracy of initialized experiments than FREE.

The Barents Sea (BS) is representative for the Atlantic Arctic regions covered by thin ice (<1m) where SIE varies with interannual variability of the Atlantic ocean heat transport and is strongly modulated by local wind patterns (Tietsche et al., 2018; Bliss et al., 2019). Figure 10 (top, left) shows that SIC in the BS benefits most from ocean initialization (AI0/FREE) among all regions, with positive scores through the ice growth to melting seasons for up to 5-6 lead years, suggesting the Atlantic heat inflow as the major source of predictability (see improved SST in Fig. S4, top, left). In summer (August and September), the SIE retreat in northern BS is more controlled by surface wind than SST, therefore, it is a challenge to predict (Dai et al., 2020). In general the ocean initialization shows some promising improvements of SIC in March (i.e. maximum SIE)

by constraining SST for up to 5 lead years in all initialized experiments (Fig. S4, top). Compared with AI0, the skill difference for SIC in AI1 is negative (i.e. crossed) in the melting season for lead years 2-6 (Fig. 10, top, 2nd left), but positive (i.e. dotted) between July and October. The degradation in AI1, accompanied by degraded skills of SST (Fig. S4, top, centre), is probably related to the combination of initialized anomalies with high-SIC (REF) and high-SIT (FREE1) for up to 5-6 years lead time (Figure 3). Compared with AI1, SIT initialization (AI2, Fig. 10, top, 3rd left) contributes to added skill for SIC in summer

535     months (June to August) and winter months (October to next March) for up to 5-6 lead years. The enhanced skills for SIC in AI2 coincide with improved winter SST in the BS up to 3-4 lead years (Fig. S4, top, right), suggesting the important role of SIT initialization in constraining the retreat of MYI extent and the atmosphere states over the CAO, namely some predictive capability for summer SIE with winter preconditioning (Holland et al., 2011; Chevallier and Salas-Mélia, 2012; Blanchard-Wrigglesworth and Bitz, 2014; Day et al., 2014). This in turn can improve the ocean circulation and local wind-driven ice

transport in the adjacent northern BS during summer. With lead times longer than 5 years, there is a tendency of larger SIE, SIV and colder TAS over in the Arctic in all AI experiments (Fig. 4). Ocean initialization (AI0) begins to lose constraints on the development of annual maximum SIE (Fig. S4, top, left, in blue in lead years 7-9). By contrast, the added skills of AI1 emerge from lead years 7 onwards (dotted). The re-emerging skill may come from the ocean decadal predictability (Dai et al., 2020), because the degradation of SIC in lead years 7-9 associated with a cold SST anomaly in AI0 is also found in AI1 and AI2

with 1-2 year time lag (not shown). The occurrence of the cold SST anomaly in this case is not predictable but randomly arise from internal variability of the coupled atmosphere and ice-ocean system, where the time lag can be attributed to the altered atmosphere-ocean heat exchange by corrected SIC (AI1) and SIT (AI2), respectively. With respect to SIT, there is almost no improvement from all experiments in the BS (Fig. 10, bottom, left three). The summer SIC bias (Fig. S2c) seems to play a dominant role in summer SIT prediction so that AI0, AI1 and AI2 show negative skill scores between July to November in

most years. For the first 5 lead years, AI1 results in the worst prediction for SIT (compared to FREE and AI0 indicated in blue and crosses, respectively), in contrast to an opposite effect from SIT initialization in AI2.

       The GIN Seas, as a neighboring sea to BS, is the second most beneficial region for SIC (Fig. 10, top, right three) showing a similar impact of ocean initialization in AI0 with improved skills from July to next February for the first few lead years. However, the prediction skill is degraded for longer lead times. There is a contradicting feature of negative skills for winter SST

in all initialized experiments compared to FREE (Fig. S4, upper 2nd rows), presumably due to the dominant effect of winter SIC bias along the sea ice edge (Fig. S2a). As mentioned before, the variability of sea ice in the MIZ is largely influenced by the oceanic heat flux convergence and the benefit from local sea ice initialization do not survive beyond the first summer. Therefore, there is less benefit from SIC initialization to SIC prediction (only present in year 1 between April to September) than that from SIT initialization prevailing from October to next February for several lead years (dotted in Fig. 10, top, right

two). The benefit from SIT initialization on multi-year time-scales may originate from remote regions (CAO-MYI) as GIN is strongly influenced by the Arctic outflow along the EGC with MYI export. With respect to SIT prediction, the area with SIT bias in winter is larger than that in summer (Fig. S2b and d). Consequently improved SITs in all experiments relative to FREE are only prominent in summer months between July and September (Fig. 10, bottom, right three). AI2 outperforms the other

initialization methods in reducing forecast errors of SIT almost for all lead years in GIN, particularly in summer, GIN thus being the region benefitting the most from SIT initialization.

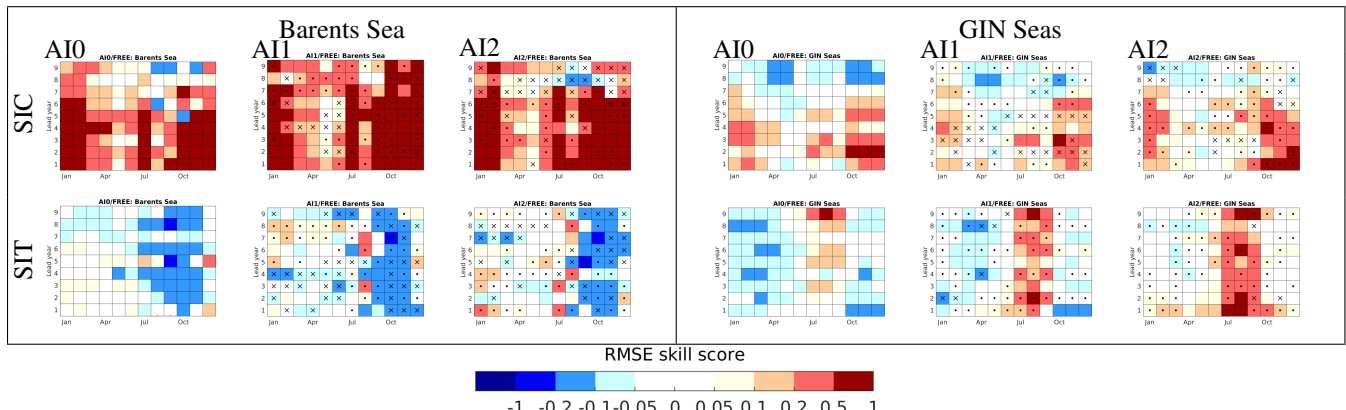

**Figure 10.** Regional Arctic SIC (upper) and SIT (lower) prediction in the Atlantic Arctic sector, i.e. Barents Sea (left) and GIN Seas (right): RMSE skill score of AI0/FREE, AI1/FREE and AI2/FREE, respectively. AI0/FREE is calculated as 1-(RMSE$_{AI0}$/RMSE$_{FREE}$), where the ratio of RMSE is averaged over regions. Scores above 0 denote more accurate in AI0 than FREE (red), and vice versa (blue). White colors denote 0 score, which means RMSEs in AI0 (or AI1, AI2) and FREE are equal, respectively. Boxes for AI1/FREE and AI2/FREE are stippled by dots (or crosses) if the added skill is above 0.05 (or below -0.05), which is the minimal color interval of RMSS skill score. The added skill is calculated for AI1 as (AI1/FREE - AI0/FREE) and for AI2 as (AI2/FREE - AI1/FREE), respectively.

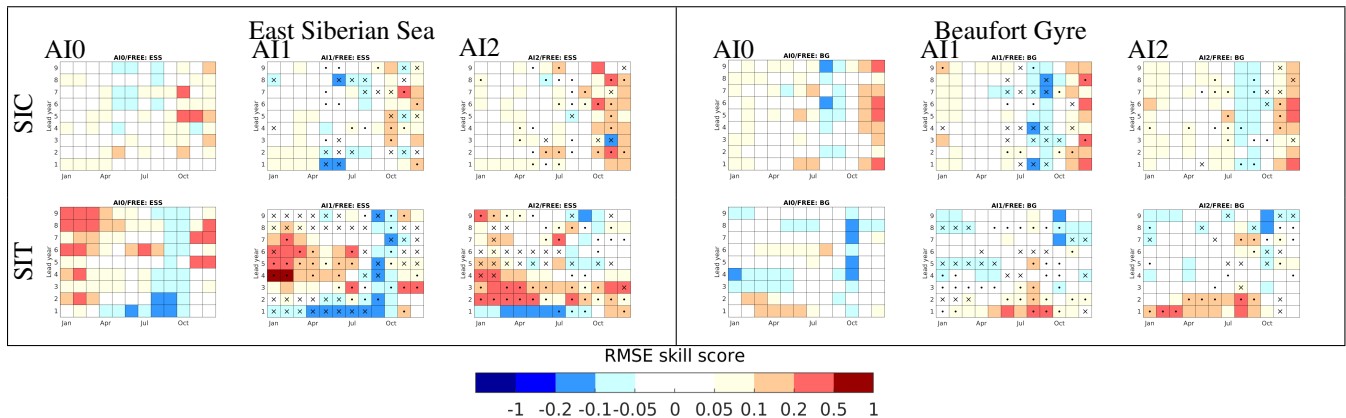

**Figure 11.** As Fig. 10, but for the Pacific shelf seas: East Siberian Sea (left) and Beaufort Gyre (right).

In the Pacific Arctic, represented by the ESS, the roles of the Pacific heat inflow in summer and the Siberian High in the ice-growth months are more important than the Atlantic inflow. Both the reanalysis (i.e. ORAS5, Tietsche et al., 2018) and the FREE simulations (Fig. S2b and d) tend to produce too thick ice in winter and too slow melting in summer. Therefore, there is little improvements in SIC prediction in all experiments when the region is fully ice-covered until March (i.e. no trend and

no interannual variability in maximum SIE). Comparing AI1 with AI0 (Fig. 11, left two), both SIC and SIT predictions are degraded by SIC initialization in the first two lead years, and slightly improved in lead years 4-6. By contrast, AI2 outperforms AI0 and AI1 in advancing the prediction skills of winter SIC and SIT with a lag time of 12 months. It corroborates with the finding in the RMSESS maps with no skill in first winter but high skill in years 2-9 (see AI2/AI0 in Fig. 6 versus Fig. 9). The SIT predictability may arise from remote origins driven by advective processes or winds (Guemas et al., 2016), because ICs of

SIC and SIT are identical in AI1 and AI2 in the ESS (see Fig. 2b). The lag time of of emerging skill between SIC (or SIT) and SST in AI0 suggests some slower adjustments between ocean initial conditions, thick ice and atmosphere than other regions.

    The sea ice cover in the Beaufort Sea is dominated by a clockwise drift, i.e. BG. It is fully ice-covered in winter and with a decreasing trend in summer SIE with higher interannual variability of thick ice cover than thinner ice cover (Bliss et al., 2019). However, there are substantial positive biases in September SIC and SIT (Fig. S2c and d). Similar to the ESS, ocean

initialization in all experiments improves early winter SIC (November and December) in some years (Fig. 11, top). The SIC and SIT initializations improve the skill in SIT in the first 3 lead years, respectively. This confirms the discussion in section 4.1 (Fig. 6 and 9, centre) that the immediate benefit from local sea ice initialization cannot hold on longer lead time up to a decade.

## 5   Summary and conclusions

This study addresses the following questions using the global climate model EC-Earth3: can sea ice initialization improve

the Arctic decadal prediction skill? Where and when may the prediction of regional seas benefit/degrade from SIC and SIT initialization? Three predictability regimes are classified according to added skill by ocean (AI0-FREE), SIC (AI1-AI0) and SIT (AI2-AI1) anomalies in the initialization:

- In general ocean initialization included in all three initialization strategies is capable of increasing prediction skill for winter SIC for a decade, thus known for the most important source of decadal predictability. In the Atlantic sector

with predominant FYI, variability and trend of SIE is largely influenced by oceanic heat flux in winter. Therefore, the errors in SIC prediction are greatly reduced by initialized ocean temperature anomalies in the first winter (Fig. 6). Many global climate models have shown that the upper ocean heat content significantly contributes to the prediction skill of sea ice in the MIZ (Bushuk et al., 2019; Cruz-García et al., 2019; Dai et al., 2020). Therefore, little improvement from sea ice initialization can be expected in these regions. Consistent with their results, we found that there is significant

degradation of SIC skill in the Hudson Bay in the first winter prediction in both AI1 and AI2. Among all Arctic regions, the BS has highest RMSE skill scores in AI0 for up to 5-6 years during the melt-to-growth seasons (July to next February in Fig. 10). This is likely attributable to the persistence of SST anomalies and advection of ocean temperature anomalies from the North Atlantic. The ACC for years 2-9 shows that there is significant skill of SIC (AI0) outskirt of the CAO-MYI region (Fig. 8), coincided with the significant skill of (TAS) over the regions and expanding over land. This reflects

the increasing impact of ocean heat inflows and local ocean-atmosphere heat exchange on the polarward retreat of SIE at decadal time-scale. All hindcasts show generally higher skill of SIC in the Atlantic sector than in the Pacific sector, consistent with the results in Dai et al. (2020).

- In comparison with AI0, AI1 is initialized with higher SIC anomalies (from REF) for up to 5-6 years lead time (Figure 3) but SIT ICs both from FREE1. As SIV is calculated as a product of SIC and SIT, only correcting SIC will result in inconsistent initial SIV fields, affecting the forecast errors of sea ice through the melting season and thereby degrade prediction skill of SIE (Blockley and Peterson, 2018; Kimmritz et al., 2018). There are some evidences of degradation (i.e. AI1-AI0) in SIC/SIT/SST in the Atlantic Sector (BS/KS/GIN) in Fig. 10 and Fig. S4-7 coincidentally in the first half year for up to 5-6 years lead time. On the other hand, a negative bias of SIC (Fig. S2a) in the Bering Sea could compensate the excessive SIC added by AI1, so there are added skills in SIC/SIT/SST in the first winter prediction (up to May in Fig. S5-7). By contrast, the Atlantic Sector shows an opposite effect due to the positive SIC bias. Alternatively, assimilating SST is recommended to effectively constrain the development of model bias.

- AI2 (with initialized SIT) outperforms AI0 and AI1 in best constraining the forecast errors of the total Artic SIV in the first 12 prediction months (Fig. 7) and in increasing correlation of SIC in the Arctic shelf seas in years 2-9 average prediction (Fig. 8). This corroborates the findings by Cruz-García et al. (2019) that the central Arctic SIV and the pan-Arctic SIE are correlated in September over three continuous years in all six global climate models used in their "idealized" experiments. Theoretically, the variability of thick ice has little connection with the upper ocean, due to the insulating role played by the sea ice cover during most of the year (Flato, 1995). From observations, the variability and trend of perennial SIE are found to be the largest in the September SIE minimum, in contrast to seasonal SIE which characterizes as ice-free in summer and by the largest variability and trend in winter (Onarheim et al., 2018). Our results provide some evidences that SIT in the melting season can be improved by constraining SIV anomaly in the CAO-MYI. One example is the GIN Sea, where there are large model biases in SIC and SIT in March and September, due to the strong impact of sea ice export from the CAO along the EGC. AI2 (with SIT initialization) shows some improvement along the ice edge (i.e. SIC) relative to AI0 and AI1 along the EGC for a decade and the reduced errors in SIT along the EGC can be linked to the reduced errors in the CAO-MYI in Fig. 9. Another example is the Pacific Arctic, where sea ice is fully covered by thick ice in winter. The prediction skill of SIC in the melting season is less dependent on the ocean heat transport thus less predictable than in the Atlantic shelf seas. Instead, a thinner and weaker sea ice in the melting season is prone to be driven by wind and increase local atmosphere–ocean surface heat flux, hence determining further evolution of the sea ice cover. Our results show SIT initialization in winter can reduce SIT error in the melting season in both BG and ESS for a few lead years (Fig. 11). SIT initialization seems a promising strategy to constrain SIV (or SIT) of CAO-MYI, which in turn constrains the expansion of polarwards sea ice retreat in the melting season.

Overall, the impact of sea ice initialization on reducing sea ice forecast errors is not just limited to the first few years locally, but can also reemerge after 5-7 forecast years, suggesting prominent modes of internal variability, such as the winter NAO, Atlantic meridional overturning circulation or the variability of Arctic outflow(Proshutinsky and Johnson, 1997; Swart et al., 2015; Armitage et al., 2020). The impact of sea ice initialization contributes to 1-2 year time lag of prediction skill, due to the altered atmosphere-ocean heat exchange by corrected SIC/SIT in AI1 and AI2.

A limit of SIT from ORAS5 is that it has no constraint by SIT observations. Although we only initialize SIT in November, in which ORAS5-SIT has shown some reliable results in representing thin ice during the freezing season (October–December) over the pan-Arctic when compared with observations by Tietsche et al. (2018). Xie et al. (2018) by directly assimilating SIT show that the perennial (thick) ice in CAO could be remarkably corrected if compared with the SIC assimilation. Therefore, the present study may underestimate the impact of SIT initialization to CAO-MYI on the Arctic climate prediction skill and further investigation is needed for an accurate assessment.

To conclude, our sensitivity experiments with EC-Earth3-CPSAI by imposing different initialized model components demonstrate that AI2 (all init.) yields an improved performance for decadal prediction for the Arctic regions, as it provides an improvement in predicting SIE and SIV anomalies and reducing errors in regional sea ice states. As climate warming continues, the central Arctic that is covered mostly by MYI will likely become seasonal ice free in the future. The controlling mechanism for decadal predictability in the region may thus shift from the current SIV persistence dominated regime to a more ocean-related processes. These findings state the foundation for the AI2-approach being the choice for a full contribution to CMIP6-DCPP covering 60 initializations (1960-2019) with 15 ensemble members each. A more general assessment of this system's predictive skill beyond the Arctic is currently in preparation.

*Code and data availability.* The EC-Earth model (version 3.3.1.1) with its standard coupled model configuration (T255L91-ORCA1L75) is used for the experiments here. The entire code of EC-Earth is not available due to restrictions in the distribution of the atmosphere component IFS. Confidential access to the entire code can be granted for editors and reviewers; please use the contact form at http://www.ec-earth.org/about/contact. For the methods of anomaly initialization to ocean and sea ice, we followed the approach described by Hazeleger et al. (2013), namely by adding reanalysis anomaly to model climatology; this can be implemented at one command line with the utility of Climate Data Operator (CDO). The programmes used to convert one-category sea ice initial states (i.e. SIC, SIT and SNT) to 5 categories and the scripts used to produce the figures are available at https://doi.org/10.5281/zenodo.4297603 (Tian et al., 2020). Data used in this paper are available at https://doi.org/10.5281/zenodo.4297926 (Tian, 2020). Links to model output of sensitivity experiments can be found from the aforementioned URL. The CMIP6 data (e.g. FREE and AI2) can also be downloaded from any Earth System Grid Federation (ESGF) data portal.

*Author contributions.* TT conceived the idea of sensitivity study, performed the analysis and wrote the manuscript. TT and SY designed and ran the experiments. TT, SY, MPK, FM and ToK contributed to develop anomaly initialization method with EC-Earth3 for CMIP6 DCPP. All authors have substantially contributed by providing suggestions and comments to refine the story of the paper.

*Competing interests.* There is no competing interests are present.

*Acknowledgements.* We thank Magdalena Balmaseda for sharing daily data from ORAS5 reanalysis. We thank Mihaela Caian and Klaus
Wyser for their early contributions to initialized CMIP5 decadal predictions with EC-Earth V2.3, which provided the basis for this work.
We also thank Clement Rousset for his help for fixing bugs in the sea ice model. TT, SY, MPK and ToK acknowledge support from the
NordForsk research programme Arctic Climate Predictions: Pathways to Resilient, Sustainable Societies (ARCPATH No.76654). TT and SY
are also supported by the Blue-Action project (European Union's Horizon 2020 research and innovation programme under grant agreement
No.727852) and the Danish National Center for Climate Research (NCKF). SY and TiK acknowledge funding by the EU's Horizon 2020
project EUCP (Grant Agreement No.776613) and TiK also acknowledges the EU's Horizon 2020 projects INTAROS and AfriCultures (Grant
Agreements No.727890 and No.774652). The EC-Earth3 simulations at SMHI have been performed on resources provided by the Swedish
National Infrastructure for Computing (SNIC) at the National Supercomputer Centre at Linköping University (NSC).

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
