# Peer review of "Benefits of sea ice initialization for the interannual-to-decadal climate prediction skill in the Arctic in EC-Earth3"

_Geoscientific Model Development, 2020_

## Referee Comment (RC1) · Anonymous Referee #1 · 13 Feb 2021

General comments:

The authors introduce a method to perform anomaly initialization of predictions with EC-Earth3. A mapping from single-category to multi-category sea ice states has been successfully developed and used, which is useful whenever data to initialize sea ice states are not given in the multi-category framework. It is common to initialize prediction systems using reanalysis products derived from different models. Performing "anomaly initialization" has proven beneficial in earlier works. Using only a subset of variables to "(anomaly-) initialize" allows to study the origin of prediction skills, here done for sea ice in the Arctic region. This discussion provides an important contribution

to assess the contribution of particular variables, like sea ice thickness or sea ice concentration, for the prediction skill on time-scales up to decades, and has the potential to impact future decisions on initialization strategies. Due to the number of performed experiments, the material to be discussed is a lot, and the authors already confined to discuss the sea ice state and surface air temperature. It could help the improve understanding of the results and condense the information when the authors would cluster regions of similar physical properties and discuss the added/reduced skill for the different setups for these different regions. It would further improve the manuscript if the reader would be better guided through the text as done already by motivation the choices of studied variables or time-frames.

Specific major comments:

1. assessment of the results

1.1. You can better guide the reader through the script by introducing and motivating what you do in the different sections and subsections.

1.2. I suggest to restructure the assessments in Sections 3 and 4 focusing on particular regions dependent on specific physical regimes, e.g. North Atlantic section, Pacific section, Central Arctic and FYI coastal areas (as similarly done for the discussions of different regions in Sec. 4.2). Thus, one could discuss the effect of the different initializations for these regions for each experiment in a more condensed way. It would also prevent discussing some detail-regions/cases (e.g. E.Siberian Sea in l.373) while skipping others (e.g. GIN Seas for AI0 in Sec.3.3.1 or skill of SIC I FREE in l.370, or l.392ff).

1.3. The authors study the benefits of keeping certain model variables unchanged while initializing the anomalies of others. I think, this is a nice and useful approach. Could you, in the discussion section, stronger indicate adjusting which anomalies lead to which skill? I.e. stronger emphasize the role of the ocean, SIC and SIT; and indicate adding of which information reduces the prediction skill where.

2. Observed anomalies/observed state: These terms are misleading. You use fields from a reanalysis product (produced by LIM2 and NEMO3.4 with assimilated SIC, T,S, SSHA observations using 3DVar FGAT). In particular, a reanalysis-SIT, which you use as "observation". The approach introduced as in the manuscript would not be suitable to handle observational errors and data sparsity. Using the reanalysis ORAS5 as observations, you make use of the strengths of 3DVar. Please check the manuscript and correct/clarify that phrasing. Examples are: l.45ff, l.54ff, l.114

3. Prediction systems require an initial state. Here, this has been achieved by using a EC_Earth3 spinup state where some variables have been anomaly-corrected using ORAS5 reanalysis data and a fullfield ERAI state for the atmosphere. You do not apply data assimilation. Please modify the manuscript (whenever your approach is called "assimilation" or you compare your approach with the assimilation approach) accordingly. Examples are l.72, l.202. It has not been mentioned in the introduction that and why you performed fullfield initialization in the atmosphere (as mentioned in l.178). Please add according statements.

4. Non-consistent initialization: There are systems like NorCPM, in which the model starts its prediction by using a reanalysis for initialisation that stems from performing DA in their own system, while others like the GFDL system use products from different models to initialize different model components. As model physics differ, initialization shocks are likely (see for instance the EC-Earth model-based study: https://link.springer.com/article/10.1007/s00382-020-05560-4), that impact the prediction skill. You should also address this when you discuss (atm) fullfield initialization and (ocean+sea ice)anomaly initialization. You already do the latter (e.g. in l.119), yet rather indirectly.

5. Variables to be discussed: The paper aims to enhance the understanding of decadal predictability of Arctic sea ice by initializing anomalies of different subsets of variables from the ocean and the sea ice state.

5.1. The relevance of the atmosphere in the Arctic region (regarding aspects of predictability and/or necessity for skillfull predictions) should be addressed, in particular the need to study TAS. Suitable places in the text may be the introduction or Section 2.3.

5.2. A discussion on the atmosphere is lacking in the discussion section. If there is no particular need to study TAS (for instance as feedback from altered ocean/sea ice state and potentially back to the ocean/sea ice state in the Arctic region), it may be an idea to skip the discussion on TAS.

5.3. It should be addressed, why the skill in the upper ocean state in the Arctic region is not discussed, though the authors could use the ORAS5 reanalysis product (as is done for SIT) and TAS is studied. Insights might be gained for instance on the degradation in SIT in the Atlantic water inflow region for AI2 (e.g. also by studying the bias of SST or T in the upper ocean in Fig.2, which I suggest to add). One argument could be that TAS is a reflection of SST and in addition provides information on how the skill changed over land dependent on the initialization scheme.

5.4. Table1 (l.103): Snow thickness anomaly is also initialized in AI2. The role of snow cover for prediction should be discussed in the introduction. As well leave a note why you do not assess the skill for SNT, or if you did, provide a brief summary of the results, e.g. in the discussion.

Specific minor comments:

6. choice of title: The authors primarily discuss the added benefit from sea ice initialization for decadal prediction skill in the Arctic. Discussions on the improvement of AI2 over AI1 are of rather subordinated relevance. Line 71f hints towards a more suitable adaptation of the title: "benefits of sea ice initialization".

7. l.110ff: could you add a statement about the sea ice state in ORAS5? Does it compare well against observations?

8. L.143: Could you explain why Aˆctrl has only one dimension (i.e. one value per time)?

9. L. 153: "Different from weight [...], Fig.1 plots the h_lˆctrl -Aˆctrl histogram": Eq. (2) is not a plotted histogram. Please, rephrase how equation 2 is linked to Fig.1.

10. L.159: If Vˆice is not "ice volume per unit area" with unit m (compare l.18), then area should be taken into account. Otherwise, [Aˆice] should be [0,1] and not [%] (l.154). Compare also with Fig.3.

11. l.159/l.161: Could you clarify more in detail how you constructed h_lˆice? The thickness classes are defined by lower and upper bounds of each thickness bins. Do you reduce these bins to its mean value dependent on the value of Aˆice? I.e. for a given value Aˆice, are h_lˆice the exact mean values taken from Fig.1 dependent on in which bin Aˆice falls?

12. On pages 6f you explained how you constructed fullfield multicategory Aˆice, Vˆice and SITˆice fields. Could you make it clearer in l.125 that you start already with the initialized (i.e. corrected) Aˆice, along the line: "To derive anomaly-corrected Aˆice values we add the anomalies of ORAS5 ice concentrations to the climatology of AˆFREE and then split this corrected field into different thickness categories"? It is unclear to me, how you anomaly-initialize SIT. Could you add a sentence in l.161?

13. L.405, Fig 9: could you explain the degradation found in TAS over Barents and Kara Seas by added SIT initialization?

14. L.406ff (last paragraph) I do not see that clear improvement as the authors see between lead years 6-9 and 2-5 in the North Atlantic. (S5 and S7) There are mostly dipole patterns of improvement /degradation in each graph and discussed region. I would skip that part and the graphs. I do not see added information of looking into these periods for the paper.

15. L.434: Using RMSESS as explanation for moderate skill in AI2 is an unfortunate

choice as RMSESS is a relative skill score.

16. Assessment of the results. Could you provide some more explanation or correct, respectively?

16.1. l.244: you indicate that the reason for the biases in the sea ice state are linked to MYI region, which seems to contradict with the results of Fig.S2. It appear to be linked to O-A heat exchange resulting in too little ice in summer, too fast freezing in autumn and too thick ice in winter.

16.2. L.295: high skill in FREE seems to be an indication that the external forcing determines the skill in contrast to internal variability. Could you add an indication for that?

16.3. In Fig.3 it would be beneficial to also plot FREE to identify the differences between FREE and REF. You use FREE-SIC and FREE-SIT in AI0, while for AI1 REF-SIC and in AI2 additionally REF-SIT, resulting in presumably different anomalies.

16.4. L.321: SIT discussion: You derive the anomalies for ocean and sea ice from the same product, that has been constructed via 3D-Var FGAT. The initial anomalies of ocean and SIT in AI2 should not be "counteracting". I am wondering if AI0/FREE has negative skill score in that region. Might the degradation be linked to model biases (see also Fig S2) that are kept unchanged by the initialization, and O-A heat exchange?

16.5. Discussion on Fig.7 (l.349ff): It appears that different measures are used to judge "similar performance" or "improved performance", e.g. compare ACC(SIV) ("much difference") vs, ACC(TAS) ("hardly any differences").

16.6. l.392ff: The center graph in Fig.9 indicates a change in SIT which might also be seen for SIC in AI0/FREE. Changes in the Arctic basin SIT are regulated by the ocean circulation such as the Beaufort Gyre and the Transpolar drift (Davis et al. (2014)). Here, it looks like the Beaufort Gyre has been impacted by ocean anomaly initialization, which might explain as well the degradation found in Fig 8 left lower two graphs.
16.7. L.423: Can you link the skill in Barents Sea to reemergence of SST and longtime benefit from ocean initialization?

16.8. l.424: Is the added skill in the Barents Sea due to ice initialization due to SIC initialization or added SIT initialization?

16.9. Fig 10: Do you have an explanation why there is this strong degradation in the Hudson Sea and the Baffin Bay?

Technical corrections:

- L.2: "sea ice volume, being a product of sea ice area/concentration (SIC) and thickness (SIT)": remove "area"

- L.6: "regional benefits"

- L.79: "this paper is structured as following s

- L.79: "the" ensemble-experiment"al" design (potentially also remove "ensemble" as you do not address the topic of ensembles in the introduction.)

- L.82: I suggest to replace "benefits of [...] at decadal scales [...] regional mean" by "benefits of [...] at decadal scales in/for different Arctic regions"

- L.87: remove "configures".

- Check use of articles, e.g. in l.87f, l.233, l.224 ( "A TAS index")

- L.91. Use of adverb: "linearly reduced"

- L.94 : remove "embedded"

- l.104ff: "This approach[...] Since ORAS4 [...] EC-EARTH2.3": I suggest to shorten these two sentences to "This approach has already been applied to initialize EC-Earth2.3 decadal predictions contributing to CMIP5."

- L.117: "Horizontally, "

- L.124: thickness for of snow. . .

- L.140 replace "l=1-L" by "l=1,...,L" or "l ∈ {1,...,L}".Âă

- l.151: Please shift the definition of hˆice_l before equation (4-5), and introduce these two equations. Currently they are not connected with the surrounding text.

- L.155: replace "suggesting" by "and" as the two parts of the sentences address two different observations from Fig.1.

- L. 169: "an one-day spin."

- L.159: Could you shift eq.4 and 5 up to where these are introduced (l.152)?Âă

- L.166. Introduce abbreviation IC (initial condition).

- Fig.1 shows h_lˆice -Aˆice histogram, conflicting with the superscript "ctrl" in l.153.

- l.153: Fig.1 does not depict a histogram, but mean ice thickness values for different SIC ranges.

- l.192: introduce SSP2-4.5

- l.202: This is not a sentence. Check after comma.

- L.204: Consider replacing "thanks to" by "using, as also with a short assimilation window one might end up in a low quality reanalysis product.

- L.210: reference of "it" is unclear

- l.246: "initial anomalies from REF": Better call it along the line "differences between anomalies of FREE and of REF". For instance, in AI2 these initial "anomalies" should be zero.

- L.291 : Could you precise that you refer to ACC(SIC) as for instance the statement is not correct for ACC(SIT).

- L.293: replace "east of the Kara Sea" by "Kara Sea" according to the definition in Fig

S2.

- L.387: (Fig 9., center, left), l.392: "Comparing the center and the lower panels"

- L.4 in caption of Fig 3: "Backwards extensions…"

- L.352: remove "in summary". I suggest to shift this and the following sentence to l.351, where you discuss SIV. Similarly, shift the then following sentences on SIE performance where you discuss SIE performance (l.350).

- Caption Fig 4, L.4: "lines plot"

- L.376f: I would suggest to phrase that the improvements in TAS follow those of improved sea ice state in FYI regions and expand over land as well.

- L.381ff: Could you add names of variable and experiment you are discussing?

- l.397: Please add a Figure reference from section 3.3.1 you are referring to.

- L. 412: "from sections 3.3 and 4.1"

- L.415: check sentence "but its impact can…."

- L.430: Could you name the initialization you are discussing (all/AI1/AI2)?

- Fig.6 -0.5 is missing in the colorbar and last sentence in the caption: correct "The regions discard SIT initialization"

- Caption of Fig 10.: "RMSE in AI2 and FREE are equal": add "AI0, respectively".

---

## Referee Comment (RC2) · Anonymous Referee #2 · 14 Mar 2021

The article of "Benefits of sea ice thickness initialization for the Arctic decadal climate prediction skill in EC-Earth3" uses different anomaly initializations of ocean, SIC and SIT in the EC-Earth3 climate prediction system from 1987-2016. These three ensemble prediction experiments investigate the hindcast prediction skills for sea ice and air temperature near surface at different time-scales. The concerned conclusions are very interesting to well understanding the initialization uncertainty for the Arctic climate prediction from seasonal to interannual variabilities. It has a close relation for this journal, but there are some factors should be more clarification before published. • All the experimental runs are no more than 10 years. So a little adjustment of this title could

be better like using "Benefits of sea ice thickness initialization for the Arctic climate prediction from seasonal to interannual in EC-Earth3" âǍć A limit of SIT from ORAS5 which has no constraint by the SIT observations. It may result in the underestimation of the SIT initialization impact on the Arctic system. In fact, some SIT assimilation work like Xie et al. (2018) shows the perennial ice in CAO could be remarkably corrected if compared with the SIC assimilation. Xie, J., Counillon, F., and Bertino, L.: Impact of assimilating a merged sea-ice thickness from CryoSat-2 and SMOS in the Arctic reanalysis, The Cryosphere, 12, 3671-3691, https://doi.org/10.5194/tc-12-3671-2018, 2018. So this limit should be clearly mentioned in the text. âǍć Line 98-103: A 25-member ensemble has been generated by the EC-Earth3, but this study only uses the 5 members of them. What reasons has been used to choose the rest 4 members excluding the FREE1? If no direct relation between these 25 members and the used 5 members, the concerned statements may be redundant for the readers. âǍć In these sensitive experiments, the derived anomaly for initialization is quite important. So as my understanding from Table 2, the initialization anomaly for sea ice on average is equivalent to the model biases shown in Fig. 2. Could you give some comments about their differences. furthermore, if comparing the stippled SIT in Fig. 2b and in Fig. 6, they look different, is it true and the reasons? âǍć Fig. 2b shows the SIT with dots covers a considerable area in the Arctic. Does it necessary to skip so wide area like in the Laptev Sea and the East Siberian Sea, or give more explanation for the threshold of 100 m? âǍć Line 314: ". . ., RMSESS>0.2 is commonly found poleward of the September sea ice edge determined by FREE (Fig. 2a), . . ." Clearly, this inferred statement lack of enough proofs to attribute to the misfits in September alone. âǍć Figure 7 show the 95% confidence lines which is quite interesting for me. Could you introduce some words how to evaluate these results? If possible, add one paragraph for this methodology in text.

Other small comments: âǍć Does the TAS (near surface air temperature) mean the air temperature at 2 m? âǍć In Table 1, the reference dataset of ORAS5 has been signed 5 for the ensemble size. As we known, the ORAS5 reanalysis assimilated the

observations in ocean and the satellite-based sea ice concentration. The related 5 members from ORAS5 are not clear, and especially for their differences. • Line 304: replace "INIT denote experiments with different initialization methods, i.e. ..." with "INIT denotes the different experiment, i.e. . . ." • Line 345: It said the polar cap domain (see Fig. 2a), but the caption of Fig. 5 also mentions the polar cap domain (north of 70N). To avoid the confuse, please keep the same concept in this study. • Checking the abbreviations are explained when they emerge at first time: ORCA at Line 94; ORAS4 at Line 106; SSP2-4.5 at Line 192;

---

## Author Response (AR1)

General comments:
The authors introduce a method to perform anomaly initialization of predictions with EC-Earth3. A mapping from single-category to multi-category sea ice states has been successfully developed and used, which is useful whenever data to initialize sea ice states are not given in the multi-category framework. It is common to initialize prediction systems using reanalysis products derived from different models. Performing "anomaly initialization" has proven beneficial in earlier works. Using only a subset of variables to "(anomaly-) initialize" allows to study the origin of prediction skills, here done for sea ice in the Arctic region. This discussion provides an important contribution to assess the contribution of particular variables, like sea ice thickness or sea ice concentration, for the prediction skill on time-scales up to decades, and has the potential to impact future decisions on initialization strategies. Due to the number of performed experiments, the material to be discussed is a lot, and the authors already confined to discuss the sea ice state and surface air temperature. It could help the improve understanding of the results and condense the information when the authors would cluster regions of similar physical properties and discuss the added/reduced skill for the different setups for these different regions. It would further improve the manuscript if the reader would be better guided through the text as done already by motivation the choices of studied variables or time-frames.

**Reply:** We thank the Referee for the comments and have revised the manuscript following these comments. Please see below our point-by-point replies. Texts with the Italic font in the answers below show the new additions to the revised manuscript.

**Specific major comments:**

1. assessment of the results
1.1. You can better guide the reader through the script by introducing and motivating what you do in the different sections and subsections.

**Reply:** We add a short introduction in Section 3 and explain why we separate prediction year 1 and years 2-9 in two sections. We reformulate the last paragraph in Section 1 on the paper structure accordingly (Line 86-89).

1.2. I suggest to restructure the assessments in Sections 3 and 4 focusing on particular regions dependent on specific physical regimes, e.g. North Atlantic section, Pacific section, Central Arctic and FYI coastal areas (as similarly done for the discussions of different regions in Sec. 4.2). Thus, one could discuss the effect of the different initializations for these regions for each experiment in a more condensed way. It would also prevent discussing some detail-regions/cases (e.g. E.Siberian Sea in l.373) while skipping others (e.g. GIN Seas for AI0 in Sec.3.3.1 or skill of SIC I FREE in l.370, or l.392ff).

**Reply:** We appreciate the reviewer's suggestions and took them into consideration as following:
  1) Add Fig.2c (incl. Figure caption, see below) and introduce it in Section 2.3 (Line 262-270).
*"According to the dominating physical regimes (Serreze and Meier, 2019), the regions studied are sorted into six groups in Fig. 2c (abbreviation explained), representing (1-3) the Arctic Ocean in the CAO ( the central Arctic 80° N north, CAA and the BG region), the Pacific Arctic (CS/ESS/LS), and the Atlantic Arctic (KS and BS); and (4-6) the MIZ in the Atlantic (HB, BAF and LAB) and in the Pacific (BER) and the transition waters between the ice-covered polar seas and the Atlantic (GIN). "*
  2) In Section 3.3.1 (Fig.5-6) and Section 4.1 (Fig.8-9), we assess two domains: CAO and MIZ.
  3) In Section 4.2, we select the Barents, GIN, E. Siberian Seas and Beaufort Gyre as representatives of the dominating physical regimes in the Atlantic/Pacific Arctic shelf seas adjacent to the CAO. Fig.10 and Fig.11 are adjusted. We explain the purpose in Line 512-519.
*" In the final analysis, we aim at providing details on the temporal evolution of the skills of initialized hindcasts for the Atlantic/Pacific Arctic shelf seas adjacent to the CAO, with evidence of the expansion of polarward retreating sea ice (Bliss et al., 2019). As the climate warming continues, the CAO might become seasonally ice free in the future. Our goal is to shed light on how the key mechanism governs the decadal predictability of the Arctic sea ice in the coming decades. Therefore, the Barents, GIN, E. Siberian Seas and Beaufort Gyre are selected to represent respective physically-dominated regimes. The FYI regions (HB/BAF/LAB/BER) are of little interest here, as they are heavily influenced by Atlantic/Pacific Ocean dynamics (also system errors) and the role of CAO-MYI diminishes. The regional assessments with groups (Fig. 2c) are shown in Fig. S5-S7."*

(c)

[Figure]

**"*Figure 2c.* (c) Regions considered for Arctic sea ice. Atlantic MIZ (brown): Hudson Bay (HB), Baffin Bay (BAF) and Labrador Sea (LAB); The Greenland/Iceland/Norwegian Seas (GIN, red); Atlantic Arctic (pink): Barents Sea (BS) and Kara Sea (KS); Central Arctic Ocean (CAO, purple): central Arctic (CAR, 80◦ N north), Canadian Archipelago (CAA), and Beaufort Gyre (BG); Pacific Arctic (blue): Laptev Sea (LS), East Siberian Sea (ESS), and Chukchi Sea (CS); Bering Sea (BER, green). "**

1.3. The authors study the benefits of keeping certain model variables unchanged while initializing the anomalies of others. I think, this is a nice and useful approach. Could you, in the discussion section, stronger indicate adjusting which anomalies lead to which skill? I.e. stronger emphasize the role of the ocean, SIC and SIT; and indicate adding of which information reduces the prediction skill where.

**Reply:** We now add AI1 in Section 4.1 (Fig. 8-9) and 4.2 (Fig. 10-11) and explain/discuss the difference with AI0 and AI2 (Line ?). We update abstract and conclusion accordingly (Line?). We first introduce the purpose in section 2.2 (Line229-232).
" *The benefit from ocean initialization (AI0) is known for the Arctic MIZ (Volpi et al., 2017; Dai et al., 2020), hence we will not address it here. Instead AI0 is used to compare with AI1 to assess the added skill with SIC initialization, and the difference between AI2 and AI1 is used to evaluate the skill gained by SIT initialization.*"

2. Observed anomalies/observed state: These terms are misleading. You use fields from a reanalysis product (produced by LIM2 and NEMO3.4 with assimilated SIC, T,S, SSHA observations using 3DVar FGAT). In particular, a reanalysis-SIT, which you use as "observation". The approach introduced as in the manuscript would not be suitable to handle observational errors and data sparsity. Using the reanalysis ORAS5 as observations, you make use of the strengths of 3DVar. Please check the manuscript and correct/clarify that phrasing. Examples are: l.45ff, l.54ff, l.114

**Reply:** We agree with the reviewer and changed "observed" to "reanalysis" throughout the text in the revised manuscript ( Line 59, 63, 75, 129).

3. Prediction systems require an initial state. Here, this has been achieved by using a EC_Earth3 spinup state where some variables have been anomaly-corrected using ORAS5 reanalysis data and a fullfield ERAI state for the atmosphere. You do not apply data assimilation. Please modify the manuscript (whenever your approach is called "assimilation" or you compare your approach with the assimilation approach) accordingly. Examples are l.72, l.202. It has not been mentioned in the introduction that and why you performed fullfield initialization in the atmosphere (as mentioned in l.178). Please add according statements.

**Reply:** We clear the term "Assimilation" in the revised manuscript (Line 79).

We rephrase the original l202 to "*We note that the reference data (ORAS5 and ERAI) have been produced by assimilation of observations into NEMO and IFS and are thus not fully independent from EC-Earth3*" (in the revised MS Line 235-237).

For the choice of FFI, we added a few lines in section: 2.2 (Line 207-2108).
"*Because initial atmospheric conditions do not play an important role for interannual to decadal predictions, for convenience, ERA-Interim (hereafter as ERAI, Dee et al., 2011) is applied as atmosphere ICs for all initialized experiments with FFI . The same initialization strategy with AI to ocean and sea ice and FFI to atmosphere with identical atmosphere reanalysis to force ocean reanalysis has already been performed for the EC-Earth2.3 decadal experiments (Hazeleger et al., 2013; Volpi et al., 2017).*"

4. Non-consistent initialization: There are systems like NorCPM, in which the model starts its prediction by

using a reanalysis for initialisation that stems from performing DA in their own system, while others like the GFDL system use products from different models to initialize different model components. As model physics differ, initialization shocks are likely (see for instance the EC-Earth model-based study: https://link.springer.com/article/10.1007/s00382-020-05560-4), that impact the prediction skill. You should also address this when you discuss (atm) fullfield initialization and (ocean+sea ice) anomaly initialization. You already do the latter (e.g. in l.119), yet rather indirectly.

**Reply:** We add the followings:
We introduce it in the beginning of Section 3 and emphasize the purpose to identify the time scale of initialization shocks and system errors attributed to forecast errors in Section 3 (Line272-280). Also see reply to RC#1-16.3.
" *It is important to measure the time-scale of ICs re-adjustment when using the first seasonal mean for seasonal prediction. In a seasonal prediction system with EC-Earth3, the forecast errors of Arctic sea ice are first attributed to the incompatibilities between the initialized variables, which causes a local dynamical readjustment in a couple of weeks, and then dominated by model inherent bias (Cruz-García et al., 2021). While in a perfect-model seasonal prediction system (no errors in ICs), the forecast errors of SIV can vary with different initialized seasons (Bushuk et al., 2019). Both studies highlight the skill gaps due to two sources of forecast errors. In this section, we characterize the initialized anomaly of sea ice states and the spatial pattern of system errors. We attempt to answer: 1) whether initial forecast errors due to the incompatibility between initialized model variables prevails over one season or a year; 2) after the model drift due to model bias becomes prominent within a decade, would the prediction years 2-9 be representative for decadal prediction in the section 4?*"

The section 3.2 Forecast drift in the original manuscript was used to address the second question. Now we add our answer to the first question in Section 3.2 Forecast drift (Line 330).
" *The roles of initialization shocks are not evident in all AIs since the first prediction year (from January 1). It is consistent with the results of Cruz-García et al. (2021) that the readjustment between surface ICs takes place within the first few weeks.*"

5. Variables to be discussed: The paper aims to enhance the understanding of decadal predictability of Arctic sea ice by initializing anomalies of different subsets of variables from the ocean and the sea ice state.

5.1. The relevance of the atmosphere in the Arctic region (regarding aspects of predictability and/or necessity for skillfull predictions) should be addressed, in particular the need to study TAS. Suitable places in the text may be the introduction or Section 2.3.

**Reply:** We agree with the reviewer's comment.
- We added few lines in the Introduction (Line 70-72).
" *In the Barents Sea (BS), northerly wind anomalies can increase sea ice export from the CAO to the BS and reduce the Atlantic inflow through the BS Opening (Dai et al., 2020). It in turn will alter local TAS via ocean and atmosphere heat exchange.*"

- We also explained the reason to study TAS, instead of SST in Section 2.3 (Line 254, w.r.t reply to 5.3)
"*Additionally, TAS over the Arctic is assessed in order to identify the local response to anomaly-corrected SST/SIC/SIT at different time scales as well as the impact over land due to different initialization schemes. A TAS index is computed as ....*
*The skill of SST is closely related to that of TAS which is representative for the ocean-atmosphere heat exchange in the open water. We include SST in supplement (Fig.S4 and S7) to support the regional skill assessment, when comparing the emerging/degradation of skill in SIC and SIT dependent on initialization scheme.*"

5.2. A discussion on the atmosphere is lacking in the discussion section. If there is no particular need to study TAS (for instance as feedback from altered ocean/sea ice state and potentially back to the ocean/sea ice state in the Arctic region), it may be an idea to skip the discussion on TAS.

**Reply:** We add some statements throughout the text to this respect.
- We introduce the purpose of analyzing TAS in Line 346. " *As mentioned before, the objective of assessing TAS changes is to identify local changes in ocean-atmosphere heat exchange altered by sea ice initialization.* "

- We explain and discuss TAS in section 3.3 and section 4 as follows:
  Fig. 5 (Line 358 and 362): " *In the MIZ (Fig. 5), FREE generally shows no significant skill (i.e. low correlation) in SIC and SIT, except some parts of GIN/BS/KS. Coincidentally over the whole regions of GIN/BS/KS, FREE shows high skill in TAS, reflecting the influence of the increasing Atlantic heat inflow since 1990s (Serreze and Meier, 2019).* " ..." *Comparing AI2 to AI1, the SIT initialization significantly enhances the high correlation areas for SIC, SIT and TAS in some parts of the Baffin Bay and KS. The major benefits of AI2 (TAS) are seen outside of the polar cap domain, manifested as significantly enhanced correlations over the North Atlantic Ocean.* "

  Fig. 6 (Line 383-388): " *By linking the skill changes in the local TAS, the degradation of SIT in the ESS/LS is probably attributed to advection of corrected SIT from the Chukchi Sea driven by local wind pattern, which prevails over external forcing, while the improved skill (SIT) in KS may originate from its neighboring waters (CAO/BS) with corrected SIT. FREE is best for TAS in large parts of the CAO with respect to both ACC and RMSE, presumable because the atmospheric large scale circulation in all initialized experiments is undertaking adjustment to the initialized states in the first few months.* "

  Fig. 7 (Line 445-449): " *Interestingly, the confidence intervals of all experiments mostly go negative between February and August (marked by open circles), indicating insignificant skill during the melting season, especially AI0. It can be linked to the overestimated thick ice cover by September in the Arctic shelf seas (Fig.S2c and d) that prevent rapid heat exchange between atmosphere and the upper ocean during the melting season. Therefore, it is challenging to constrain the atmospheric states in the first 12 months forecast with sea ice anomaly initialization.* "

  Fig. 9 (Line 493-507): " *For TAS the area with improved skill in AI2/FREE (Fig. 9, right) is considerable related to increasing ocean temperature of the Atlantic/Pacific inflows (Serreze and Meier, 2019), covering the N. Atlantic, the eastern Arctic and expanded over land. The added skill in AI2/AI1 in the landward vicinity of the BS/ESS may result from changes in local wind pattern due to the thinning of SIT in AI2 relative to AI1. Interestingly, TAS over the KS is degraded in AI2/FREE and AI2/AI0 (but not in AI2/AI1), indicating a negative effect from both ocean temperature and SIC initialization (AI0 and AI1). We should note that there are remarkable warm biases of the annual mean in SST and TAS for the BS/KS (see FREE1 in Fig. S3a and c), accompanied with a large retreat of summer sea ice in FREE1. It suggests the warm bias of SIC in summer prevails over the cold bias of SIC in winter (Fig. S2a and c) in years 2-9 average prediction. Although we have found the dominant role of ocean initialization in improving SIC and TAS prediction over the BS/KS in winter (Fig. 6), it seems to have an opposite effect in summer. Dai et al. (2020) have shown that the seasonal prediction skill for the September SIE in the BS cannot be gained by assimilating SST (or correcting SIC) alone due to lack of constraint on surface winds. Consistent with their results, SIT initialization seems promising to constrain the summer retreat of MYI (i.e. thick ice) in the northern BS/KS, thus advancing or blocking the Atlantic heat inflow to the KS. By contrast, the cold bias of winter SIC prevailing over that of summer SIC in the Labrador Sea (Fig. S2 versus Fig. S3), so that corrected (thinner) SIT will make the excessive sea ice cover (biased SIC) easier to be driven by winds. In turn, the skill in TAS (via ocean and atmosphere heat exchange) will be improved.* "

  Fig. 10 (Line 545): " *The occurrence of a cold SST anomaly in this case is not predictable but randomly arise from internal variability of the coupled atmosphere and ice-ocean system, where the time lag can be attributed to the altered atmosphere-ocean heat exchange by corrected SIC (AI1) and SIT (AI2), respectively.* "

5.3. It should be addressed, why the skill in the upper ocean state in the Arctic region is not discussed, though the authors could use the ORAS5 reanalysis product (as is done for SIT) and TAS is studied. Insights might be gained for instance on the degradation in SIT in the Atlantic water inflow region for AI2 (e.g. also by studying the bias of SST or T in the upper ocean in Fig.2, which I suggest to add). One argument could be that TAS is a reflection of SST and in addition provides information on how the skill changed over land dependent on the initialization scheme.

**Reply:** Indeed, our argument was that TAS is a reflection of SST.
- We prefer to not include the SST bias in Fig.2, because it is only prominent in the Arctic marginal ice zone, which are better reflected by the SIC bias in Fig. 2. We include plots of SST biases as supporting figures to the TAS bias in Fig.S3. As it is expected, TAS bias in the open water mirrors

SST bias in the Pacific and Atlantic Ocean.
- We explain the motivation to study TAS in Section 2.3 (Line 254). Same as reply to 5.1.
- We add the regional skill assessment of SST in Fig. S4, which is associated with the discussion about the emerging/degradation of skill in AI2 in Section 4.2.

5.4. Table1 (l.103): Snow thickness anomaly is also initialized in AI2. The role of snow cover for prediction should be discussed in the introduction. As well leave a note why you do not assess the skill for SNT, or if you did, provide a brief summary of the results, e.g. in the discussion.

**Reply:** We add one sentence to Introduction (Line 38). "*Snow cover can affect sea ice predictability in several competing ways: in early spring, the presence of snow causes the local albedo to be high, hence delaying melt onset. However, when snow melts, it forms pools of water at the surface of sea ice known as melt ponds, with a relatively lower albedo (Schröder et al., 2014).*"

The results of SNT have not yet been included in this study. We add explanation in Section 2.3 (Line 260). "*because there are very few observations on snow depth over sea ice, leading to large uncertainties in observations and so as evaluations (Tian-Kunze et al., 2014).*"

**Specific minor comments:**

6. choice of title: The authors primarily discuss the added benefit from sea ice initialization for decadal prediction skill in the Arctic. Discussions on the improvement of AI2 over AI1 are of rather subordinated relevance. Line 71f hints towards a more suitable adaptation of the title: "benefits of sea ice initialization".

**Reply:** Yes. We take suggestions from both referees.
- New title: "*Benefits of sea ice initialization for the interannual-to-decadal climate prediction skill in the Arctic in EC-Earth3*"
- New heading: "*Section 4.2 regional-mean skill for interannual-to-decadal time scales*".

7. l.110ff: could you add a statement about the sea ice state in ORAS5? Does it compare well against observations?

**Reply:** This is added in Line 123-128.
"*In comparison with other ocean-sea ice reanalysis (Chevallier et al., 2017), ORAS5 represents Arctic sea ice reasonably well and the error in SIV (up to 10%) is comparable to the uncertainties in PIOMASS (Schweiger et al., 2011). Tietsche et al. (2018) found there is good agreement between SIT from ORAS5 and from L-band observations for thin ice in the freezing season (October–December) with respect to the interannual variability and trends of thin sea-ice area over the pan-Arctic. Therefore, it is very promising to apply ORAS5-SIT to initialize the decadal prediction, which typically starts on 1 November.*"

8. L.143: Could you explain why $\hat{A}$ctrl has only one dimension (i.e. one value per time)?

**Reply:** Yes. Indeed $A$^ctrl has three dimensions as (lon,lat,time), so does the weight function *weight* ^ctrl. In order to simplify (shorten) the formula, there was "For SIC at a specific grid point" in the context, "thus, $A$^ctrl has only time dimension". But to avoid confusion, we add x, y to equation (1-5) and Fig.1 caption, and we rephrase this in the revised manuscript (Line 158-188).

9. L. 153: "Different from weight [...], Fig.1 plots the $h\_l$^ctrl -$A$^ctrl histogram": Eq. (2) is not a plotted histogram. Please, rephrase how equation 2 is linked to Fig.1.

**Reply:** Fig.1 shows a relationship between mean thickness at each category and the total sea ice concentration, which is independent of the weight function in Eq. (2).
- In Line 162
"*We assume that based on the same EC-Earth3 model version, $g_l^{ice}(x, y)$ is likely regulated by the weighting function $weight_l^{ctrl}(x, y)$ given in Eq. (2), and determined by Eq.(3).*"
- We rephrase how eq. (4-5) is linked to Fig.1 (Line176)
"*We note that in Eq.(4) and (5), $h_l^{ice}$ does not change with geographic location and time, but depends on in which bin $A^{ice}$ falls (ranging from 0.1 to 1 at intervals of 0.1 in Fig. 1). The relationship between $h_l^{ice}$ and the total ice concentration $A^{ice}$ is derived from the 300-year control run. We assume the distribution of $h^{ice}$ on $A^{ice}$ identical in the decadal experiments and the control run. Fig. 1 shows that ...*"

10. L.159: If V^ice is not "ice volume per unit area" with unit m (compare l.18), then area should be taken into account. Otherwise, [A^ice] should be [0,1] and not [%] (l.154). Compare also with Fig.3.

**Reply:** Yes. We correct this to fraction [0,1] in revised manuscript (Line 140-146, 171-172,176-181) as well as Fig.1.

11. l.159/l.161: Could you clarify more in detail how you constructed h_l^ice? The thickness classes are defined by lower and upper bounds of each thickness bins. Do you reduce these bins to its mean value dependent on the value of A^ice? I.e. for a given value A^ice, are h_l^ice the exact mean values taken from Fig.1 dependent on in which bin A^ice falls?

**Reply:** Yes, this is correct. We provided an example in Fig.1 caption" *If A^ice falling in bin 0.1-0.2, the mean values of h_l^ice are 0.08, 0.27, 0.62,1.15 and 3.20, respectively* ". Those mean thickness h_l^ice will be redistributed by lower and upper bounds of each thickness bins in LIM3 through the spin-up initialization.

12. On pages 6f you explained how you constructed fullfield multicategory A^ice, V^ice and SIT^ice fields. Could you make it clearer in l.125 that you start already with the initialized (i.e. corrected) A^ice, along the line: "To derive anomaly-corrected A^ice values we add the anomalies of ORAS5 ice concentrations to the climatology of A^FREE and then split this corrected field into different thickness categories"? It is unclear to me, how you anomaly-initialize SIT. Could you add a sentence in l.161?

**Reply:** Yes. We have added this line about anomaly-corrected A^ice and H^ice,sn values in the revised manuscript (Line 140) and added a sentence before Eq. (4) in Line 170.
" *The initialized $V^{ice}$ at the local grid-point (x,y) is calculated as a product of the initialized $A^{ice}$ (x, y) and $H^{ice}$ (x, y) (in fraction of the grid-cell area and m, respectively), which can be splitted into each category l as in Eq.(4).*"

13. L.405, Fig 9: could you explain the degradation found in TAS over Barents and Kara Seas by added SIT initialization?

**Reply:** We found that the degradation in TAS is related to ocean and SIC initialization because there is no degradation in AI2/AI1. We provide an explanation in Line 496-505.

"*Interestingly, TAS over the KS is degraded in AI2/FREE and AI2/AI0 (but not in AI2/AI1), indicating a negative effect from both ocean temperature and SIC initialization (AI0 and AI1). We should note that there are remarkable warm biases of the annual mean in SST and TAS for the BS/KS (see FREE1 in Fig. S3a and c), accompanied with a large retreat of summer sea ice in FREE1. It suggests the warm bias of SIC in summer prevails over the cold bias of SIC in winter (Fig. S2a and c) in years 2-9 average prediction. Although we have found the dominant role of ocean initialization in improving SIC and TAS prediction over the BS/KS in winter (in Fig. 6), it seems to have an opposite effect in summer. Dai et al. (2020) have shown that the seasonal prediction skill for the September SIE in the BS cannot be gained by assimilating SST (or correcting SIC) alone due to lack of constraint on surface winds. Consistent with their results, SIT initialization seems promising to constrain the summer retreat of MYI (i.e. thick ice) in the northern BS/KS, thus advancing or blocking the Atlantic heat inflow to the KS.*"

14. L.406ff (last paragraph) I do not see that clear improvement as the authors see between lead years 6-9 and 2-5 in the North Atlantic. (S5 and S7) There are mostly dipole patterns of improvement /degradation in each graph and discussed region. I would skip that part and the graphs. I do not see added information of looking into these periods for the paper.

**Reply:** We remove this paragraph and the relevant figures in supplementary (i.e. S4-S7) and make the following adjustment:
- Section 2.3 (Line 250) "*The assessment of temporal development is performed at two separate lead times: year 1 and years 2-9.*"
- Section 4.1 (Line 455) "*prediction skill (AI0,AI1,AI2) is averaged over forecast years 2-9 and compared to the respective FREE projection.*"

15. L.434: Using RMSESS as explanation for moderate skill in AI2 is an unfortunate choice as RMSESS is a relative skill score.

**Reply:** We agree. The state of "skill" in CAO has been replaced with redefined regions in the revised manuscript.

16. Assessment of the results. Could you provide some more explanation or correct, respectively?

16.1. l.244: you indicate that the reason for the biases in the sea ice state are linked to MYI region, which seems to contradict with the results of Fig.S2. It appear to be linked to O-A heat exchange resulting in too little ice in summer, too fast freezing in autumn and too thick ice in winter.

**Reply:** We think that the model bias in the GIN and Barents Seas inside September SIE climatology can be explained by MYI as its definition. Fig. 2 and Fig. S2 do not contradict each other because they represent two different periods. Fig. 2 shows the model bias of FREE1 on November 1st over 1979-2014 used for initialization, whereas Fig. S2 shows the model bias of FREE (5-member), used for skill assessment for the later period 1997-2016.

The global warming is found accelerating since the 1990s in the EC-Earth3 CMIP6 historical ensemble (see Fig.S1b). In Fig.S1b the ensemble mean of FREE5 agrees well with the mean of all members since the 90s. FREE1 is one of the warmest ensemble members during this period, although FREE1 started from a relatively cold initial state in ensemble generation (Fig.S1a).

Fig. A (see below) shows the model bias (FREE1-REF) of SIC in March (left) and September (right) averaged over the early period 1979-1996, in contrast to Fig. S2a and Fig. S2c. The modelled climatology of September SIE (FREE1 and FREE) is outside that of ORAS5 in the GIN and Barents Seas, indicating the overestimated extent of MYI. It is consistent with the positions of September SIE climatology over 1979-2014 in Fig. 2.

However, in the FYI zone (outside of the September SIE), the positive bias of SIC in March in the Labrador/ GIN/Barents Seas disappears in September (see below), indicating the role of O-A heat exchange.

We have rephrased it in the revised manuscript (Line 286).
" *Along the ice edge of GIN (typically FYI), the positive SIC bias together with 1-2 m thicker ice suggests the role of atmosphere–ocean heat exchange resulting in too fast freezing in autumn, too thick ice in winter and too little ice in summer as inferred from Fig.S2 and S3 for the period 1997-2016.*"

[Figure]

**Figure A.** Model bias (FREE1-REF) of SIC in March (left) and September (right) averaged over 1979-1996. The mean September SIE (>15%) are shown in red/green/black for ORAS5 (REF)/FREE1/FREE.

16.2. L.295: high skill in FREE seems to be an indication that the external forcing determines the skill in contrast to internal variability. Could you add an indication for that?

**Reply:** It is because the models in general capture the trend related to the external forcing better than the internal variability. We add it in Line 355." *It indicates that its prediction skill is mostly controlled by the external forcing of anthropogenic green house gases (see in Fig.S1b) compared to internal variability (corrected via initialization) during the last two decades.*"

16.3. In Fig.3 it would be beneficial to also plot FREE to identify the differences between FREE and REF. You use FREE-SIC and FREE-SIT in AI0, while for AI1 REF-SIC and in AI2 additionally REF-SIT, resulting in presumably different anomalies.

**Reply:** We add anomalies of FREE1 to Fig.3, and explain it briefly in the revised manuscript (Line 300).

"*Figure 3 depicts different sea ice initialization strategies, e.g. in the first year low-SIC (FREE1) and high-SIT (FREE1) for AI0 in contrast to high-SIC (REF) and low-SIT (REF) for AI2. AI1 has the combination of high-SIC (REF) and high-SIT (FREE1). We note that AI0 and AI1 are initialized with different combinations of ocean (REF) and sea ice anomalies (REF or FREE1) while AI2 is not. As model physics differ, initialization shocks likely impact the prediction skill. In the meantime system errors (not corrected in anomaly initialization) may develop fast, if the REF anomalies of FYI (<10%SIC and 0.2 m SIT) are much smaller than the positive (negative) bias (> 20% SIC and 1 m SIT), such as in the Atlantic (Pacific) sector in Fig.S2. Therefore, it is essential to track the development of initialization shocks and system errors in the next section.*"

16.4. L.321: SIT discussion: You derive the anomalies for ocean and sea ice from the same product, that has been constructed via 3D-Var FGAT. The initial anomalies of ocean and SIT in AI2 should not be "counteracting". I am wondering if AI0/FREE has negative skill score in that region. Might the degradation be linked to model biases (see also Fig S2) that are kept unchanged by the initialization, and O-A heat exchange?

**Reply:** Yes. Both AI0/FREE and AI1/FREE have negative skill score for SIT in the LAB/GIN/BS region (see Fig. B, upper). The area and shape of high RMSE of SIT (AI0, AI1 and AI2, Fig. B lower, right three) are similar to the model bias of SIC and SIT in March in Fig.S2a. We add explanation in Line 390.

" *Skills (SIT) are degraded along the ice edge of the Labrador/GIN Seas in AI2/FREE (also in AI1/FREE and AI0/FREE, not shown), showing opposite skill changes with positive score (>0.5) in both AI2/AI0 and AI2/AI1 versus negative score (<-1) in AI2/FREE. The negative scores in AI2/FREE(SIT) coincide with the maximum bias of sea ice states (Fig. S2a and b), suggesting the major role of model bias in causing forecast errors.*"

[Figure]

**Figure B**. Upper row: RMSE skill score of AI0, AI1 and AI2 relative to FREE for SIT in the first winter. Low row: RMSE of FREE, AI0, AI1 and AI2 with respect to REF.

16.5. Discussion on Fig.7 (l.349ff): It appears that different measures are used to judge "similar performance" or "improved performance", e.g. compare ACC(SIV) ("much difference") vs, ACC(TAS) ("hardly any

differences").

**Reply:** as reply to RC#2-7. We make the following changes:
- We introduce how to evaluate these results in Line 420. "*Additionally, the thin lines in Fig. 7 represent the upper/lower bounds of the 95 % confidence intervals obtained with a t-distribution for correlations and a χ2 distribution for RMSE. The correlation with one experiment is not significant, if the confidence interval goes below 0. Furthermore, the difference between two experiments is not significant, if those two intervals overlap.*"
- The results in section 3.3.2 are rewritten, by adding how to interpret the 95% confidence lines in order to evaluate the skills of initialized experiments.
- For ACC (TAS) in Fig.7c, stippled points are added if correlation is insignificant (p>=0.05).

16.6. l.392ff: The center graph in Fig.9 indicates a change in SIT which might also be seen for SIC in AI0/FREE. Changes in the Arctic basin SIT are regulated by the ocean circulation such as the Beaufort Gyre and the Transpolar drift (Davis et al. (2014)). Here, it looks like the Beaufort Gyre has been impacted by ocean anomaly initialization, which might explain as well the degradation found in Fig 8 left lower two graphs.

**Reply:** We agree. There is degradation of SIC and SIT in the Beaufort Gyre (BG) region in both AI0/FREE and AI2/FREE (see Fig. C), which have been reflected by almost no difference in AI2/AI0 in Fig. 9 (lower row). This indicates the dominant impact of ocean anomaly initialization to sea ice states in the BG. We add explanation in Line 486.

"*By contrast, there are negative effects of ocean initialization with degraded skills for SIC and SIT in the BG region with negative RMSESS in AI2/FREE (also AI0/FREE, not shown) and slightly negative in AI2/AI1 in Fig. 9 (lower rows, left two). This may be associated with an increasing polarward expansion of FYI zone in the southern BG (Bliss et al., 2019), where a thinner sea ice cover (represented by local SIT anomaly in BG) will be more easily forced by wind, and consequently lead to stronger circulation in the BG (Armitage et al., 2020). However, there are substantial system bias in sea ice states in the BG in September (>20 % SIC and 2 m SIT, Fig. S2c and d), therefore, the immediate benefit from local SIT initialization (Fig. 6, centre) cannot hold on time-scales up to a decade in this respect.*"

[Figure]

**Figure C**. RMSE SS of AI0/FREE (left) and AI2/FREE (right) for SIC (upper row) and SIT (lower rows), averaged over years 2-9.

16.7. L.423: Can you link the skill in Barents Sea to reemergence of SST and longtime benefit from ocean initialization?

**Reply:** We added some explanation to Line 540.

*" With lead times longer than 5 years, there is a tendency of larger SIE, SIV and colder TAS over in the Arctic in all AI experiments (Fig. 4). Ocean initialization (AI0) begins to lose constraints on the development of annual maximum SIE (Fig. S4, top, left, in blue in lead years 7-9). By contrast, the added skills of AI1/FREE emerge from lead years 7 onwards (dotted). The re-emerging skill may come from the ocean decadal predictability (Dai et al., 2020), because the degradation of SIC in lead years 7-9 associated with a cold SST anomaly in AI0 is also found in AI1 and AI2 with 1-2 year time lag (not shown). The occurrence of the cold SST anomaly in this case is not predictable but randomly arise from internal variability of the coupled atmosphere and ice-ocean system, where the time lag can be attributed to the altered atmosphere-ocean heat exchange by corrected SIC (AI1) and SIT (AI2), respectively.*"

16.8. l.424: Is the added skill in the Barents Sea due to ice initialization due to SIC initialization or added SIT initialization?

**Reply:** It is both. The added skill in the Barents Sea is first shown in AI1 (Fig.10 with dots if the skill difference AI1/FREE-AI0/FREE>0.05) from July to October for up to 5-6 lead years. On top of it, the dots in AI2 (if the relative skill to AI1 above 0.05) are seen in summer and winter months (June-to-August and October- to-next March, respectively) for up to 5-6 lead years.

[Figure]

*"Figure 10. Regional Arctic SIC… in the Barents Sea..: RMSE skill score of AI0/FREE, AI1/FREE and AI2/FREE, respectively. AI0/FREE is calculated as 1-(RMSE$_{AI0}$ /RMSE$_{FREE}$ ), where the ratio of RMSE is averaged over regions. Scores above 0 denote more accurate in AI0 than FREE (red), and vice versa (blue). White colors denote 0 score, which means RMSEs in AI0 (or AI1, AI2) and FREE are equal, respectively. Boxes for AI1/FREE and AI2/FREE are stippled by dots (or crosses) if the added skill is above 0.05 (below-0.05), which is the minimal color interval of RMSS skill score. The added skill is calculated for AI1 as (AI1/FREE - AI0/FREE) and for AI2 as (AI2/FREE - AI1/FREE), respectively."*

We add explanation to Line 530-540.
*" Compared with AI0, the skill difference for SIC in AI1 is negative (i.e. crossed) in the melting season for lead years 2-6 (Fig. 10, top, 2nd left), but positive (i.e. dotted) between July and October. The degradation in AI1, accompanied by degraded skills of SST (Fig. S4, top, centre), is probably related to the combination of initialized anomalies with high-SIC (REF) and high-SIT (FREE1) for up to 5-6 years lead time (Figure 3). Compared with AI1, SIT initialization (AI2, Fig. 10, top, 3rd left) contributes to added skill for SIC in summer months (June to August) and winter months (October to next March) for up to 5-6 lead years. The enhanced skills for SIC in AI2 coincide with improved winter SST in the BS up to 3-4 lead years (Fig. S4, top, right), suggesting the important role of SIT initialization in constraining the retreat of MYI extent and the atmosphere states over the CAO, namely some predictive capability for summer SIE with winter preconditioning (Holland et al., 2011; Chevallier and Salas-Mélia, 2012; Blanchard-Wrigglesworth and Bitz, 2014; Day et al., 2014). This in turn can improve the ocean circulation and local wind-driven ice transport in the adjacent northern BS during summer.*"

16.9. Fig 10: Do you have an explanation why there is this strong degradation in the Hudson Sea and the Baffin Bay?

**Reply:** As reply to 1.2, the figures for two regions are removed. But we explain the reason of degradation here with Fig.D.

The two regions are located outside of the September SIE climatology and are mostly ice free in summer (white areas in Fig. D). The skill degradation is calculated based on averages over a very small area (a few points) in the eastern (southern) border of the Canadian Archipelago in the Baffin (Hudson) Bay as indicated by blue (red) circles in Fig. D.

[Figure]

**Figure D**. RMSE SS of AI0/FREE, AI1/FREE and AI2/FREE (from left to right) for September SIC in year 1, given as examples. The red and blue circles in the left figure are the sub-regions showing degradation in the Hudson and Baffin Bay, respectively.

**Technical corrections:**

17. L.2: "sea ice volume, being a product of sea ice area/concentration (SIC) and thickness (SIT)": remove "area"

**Reply:** We correct this in the revised manuscript (Line 2).

18. L.6: "regional benefits"

**Reply:** We correct this in the revised manuscript (Line 6).

19. L.79: "this paper is structured as following s

**Reply:** We correct this in the revised manuscript (Line 86).

20. L.79: "the" ensemble-experiment"al" design (potentially also remove "ensemble" as you do not address the topic of ensembles in the introduction.)

**Reply:** We agree and correct this in the revised manuscript (Line 87).

21. L.82: I suggest to replace "benefits of [...] at decadal scales [...] regional mean" by "benefits of [...] at decadal scales in/for different Arctic regions"

**Reply:** Yes. We change this in the revised manuscript (Line 89).

22. L.87: remove "configures".

**Reply:** We correct this in the revised manuscript (Line 93).

23. Check use of articles, e.g. in l.87f, l.233, l.224 ( "A TAS index")

**Reply:** We correct them in Line 92(l.87f) and Line 256 (l.224). The sentence (l.233) is removed from the revised manuscript.

24. L.91. Use of adverb: "linearly reduced"

**Reply:** We correct this in the revised manuscript (Line 97).

25. L.94 : remove "embedded"

**Reply:** We correct this in the revised manuscript (Line 98).

26. l.104ff: "This approach[. . .] Since ORAS4 [. . .] EC-EARTH2.3": I suggest to shorten these two sentences to "This approach has already been applied to initialize EC-Earth2.3 decadal predictions contributing to CMIP5."

**Reply:** We change this in the revised manuscript (Line 112).

27. L.117: "Horizontally, "

**Reply:** We correct this in the revised manuscript (Line 132).

28. L.124: thickness for of snow. . .

**Reply:** We correct this in the revised manuscript (Line 137).

29. L.140 replace "l=1-L" by "l=1,...,L" or "l ∈ {1,...,L}".Âǎ

**Reply:** We replace it with "l=1,...,L" in the revised manuscript (Line 159,179) and the caption of Fig. 1.

30. l.151: Please shift the definition of hˆice_l before equation (4-5), and introduce these two equations. Currently they are not connected with the surrounding text.
 L.159: Could you shift eq.4 and 5 up to where these are introduced (l.152)?

**Reply:** We have corrected and adjusted these two points in the revised manuscript (Line 170).

31. L.155: replace "suggesting" by "and" as the two parts of the sentences address two different observations from Fig.1.

**Reply:** We change this in the revised manuscript (Line 179).

32. L. 169: "an one-day spin."

**Reply:** We correct this in the revised manuscript (Line 192).

33. L.166. Introduce abbreviation IC (initial condition).

**Reply:** We add this in the revised manuscript (Line 88).

34. Fig.1 shows h_lˆice -Aˆice histogram, conflicting with the superscript "ctrl" in l.153.

**Reply:** We add a sentence in Line 178.
" *We assume the distribution of $h^{ice}$ on $A^{ice}$ identical in the decadal experiments and the control run.*"

35. l.153: Fig.1 does not depict a histogram, but mean ice thickness values for different SIC ranges.

**Reply:** We correct this in the revised manuscript (Line 179). As reply to RC#1-9

36. l.192: introduce SSP2-4.5

**Reply:** We add "medium" SSP2-4.5 forcing of ScenarioMIP (2015-2017, Boer et al., 2016) in the revised manuscript (Line 224).

37. l.202: This is not a sentence. Check after comma.

**Reply:** We rephrase this in the revised manuscript (Line 235).
"*We note that the reference data (ORAS5 and ERAI) have been produced by assimilation of observations into NEMO and IFS and are thus not fully independent from EC-Earth3.* "

38. L.204: Consider replacing "thanks to" by "using, as also with a short assimilation window one might end up in a low quality reanalysis product.

**Reply:** We remove the sentence because a statement about the sea ice state in ORAS5 is moved to Line 120-128 (as reply to RC#1-7).

39. L.210: reference of "it" is unclear

**Reply:** It is "*The skill assessment*". We add it in Line 241.

40. l.246: "initial anomalies from REF": Better call it along the line "differences between anomalies of FREE and of REF". For instance, in AI2 these initial "anomalies" should be zero.

**Reply:** We change this in the revised manuscript (Line 291) as well as Fig.3 caption. The anomalies of FREE1 are added to Fig.3 as suggested by RC#1-16.3.
We note that initial anomalies in AI2 should be zero when averaged over the climatological period 1979-2014 (see Line 294) " *with respect to the mean of the whole period 1979-2014 used for initialization.*" However, the 20-year averages (1997-2016) are negative in AI2.

41. L.291 : Could you precise that you refer to ACC(SIC) as for instance the statement is not correct for ACC(SIT).

**Reply:** We clarify this Line 360, also throughout the text.

42. L.293: replace "east of the Kara Sea" by "Kara Sea" according to the definition in FigS2.

**Reply:** We change it in Line 363.

43. L.387: (Fig 9., center, left), l.392: "Comparing the center and the lower panels"

**Reply:** We correct it in Line 478 and Line 480, and throughout the text.

44. L.4 in caption of Fig 3: "Backwards extensions. . ."

**Reply:** Fig.3 is modified as response to Comment 16.3. Dashed lines used to indicate "Backwards extensions" are removed from the figure.

45. L.352: remove "in summary". I suggest to shift this and the following sentence to l.351, where you discuss SIV. Similarly, shift the then following sentences on SIE performance where you discuss SIE performance (l.350).

**Reply:** We rewrite this section, by addressing SIE, SIV and TAS in separate paragraphs (Page 19-20).

46. Caption Fig 4, L.4: "lines plot"

**Reply:** We have corrected this.

47. L.376f: I would suggest to phrase that the improvements in TAS follow those of improved sea ice state in FYI regions and expand over land as well.

**Reply:** We change it in Line 467.

48. L.381ff: Could you add names of variable and experiment you are discussing?

**Reply:** Yes. We check throughout the text.

49. l.397: Please add a Figure reference from section 3.3.1 you are referring to.

**Reply:** We add "Fig.6" to the text, such as in Line 491.
" *therefore, the immediate benefit from local SIT initialization (Fig. 6, centre) cannot hold on time-scales up to a decade in this respect.*"

50. L. 412: "from sections 3.3 and 4.1"

**Reply:** We change it in Line 509.

51. L.415: check sentence "but its impact can. . .."

**Reply:** We rephrase it in Line 511.
"*However, the added skill of SIT on decadal time-scales in some FYI regions seems associated with remote regions (CAO-MYI), which may last for several forecast years with the support of accurate ocean conditions.*"

52. L.430: Could you name the initialization you are discussing (all/AI1/AI2)?

**Reply:** Yes. We check throughout the text.-

53. Fig.6 -0.5 is missing in the colorbar and last sentence in the caption: correct "The regions discard SIT initialization"

**Reply:** We a note to in the caption. "*The labels of the colorbar for skill score is asymmetric because the minimum of SS can be far below -1 in contrast to the maximum of 1.*" And we replace the last sentence in the caption with "*The stippled areas in the middle columns are the same as in Fig. 2b.*" (as reply to RC#2-4).

54. Caption of Fig 10.: "RMSE in AI2 and FREE are equal": add "AI0, respectively".

**Reply:** We have corrected this.

**Referee #2**: Interactive comment on "Benefits of sea ice thickness initialization for the Arctic decadal

climate prediction skill in EC-Earth3" by Tian Tian et al.

The article of "Benefits of sea ice thickness initialization for the Arctic decadal climate prediction skill in EC-Earth3" uses different anomaly initializations of ocean, SIC and SIT in the EC-Earth3 climate prediction system from 1987-2016. These three ensemble prediction experiments investigate the hindcast prediction skills for sea ice and air temperature near surface at different time-scales. The concerned conclusions are very interesting to well understanding the initialization uncertainty for the Arctic climate prediction from seasonal to interannual variabilities. It has a close relation for this journal, but there are some factors should be more clarification before published.

**Reply:** We thank the Referee for the comments and have revised the manuscript following these comments. Please see below our point-by-point replies. Texts with the Italic font in the answers below show the new additions to the revised manuscript.

1. All the experimental runs are no more than 10 years. So a little adjustment of this title could be better like using "Benefits of sea ice thickness initialization for the Arctic climate prediction from seasonal to interannual in EC-Earth3"

 **Reply:** We take suggestions from both referees.
- New title: " *Benefits of sea ice initialization for the interannual-to-decadal climate prediction skill in the Arctic in EC-Earth3*"
- New heading: "*Section 4.2 regional-mean skill for interannual-to-decadal time scales*".

    We'd like to add the following points, 1) our experiments follow the protocol of the CMIP6 Decadal Climate Prediction Project (DCPP): "*Initialized ensembles of predictions ....and runs are 2 months plus 10 years long ...*" in the original manuscript Line 180. 2) the first year prediction is only used to validate the imprint of initial conditions, not considered as seasonal prediction because our system is initialized with anomalies instead of data observations. 3) Our assessment focus on skill for predicting years 2-9. Therefore, we decided to call it "*interannual-to-decadal*" in the new title.

2. A limit of SIT from ORAS5 which has no constraint by the SIT observations. It may result in the underestimation of the SIT initialization impact on the Arctic system. In fact, some SIT assimilation work like Xie et al. (2018) shows the perennial ice in CAO could be remarkably corrected if compared with the SIC assimilation. Xie, J., Counillon, F., and Bertino, L.: Impact of assimilating a merged sea-ice thickness from CryoSat-2 and SMOS in the Arctic reanalysis, The Cryosphere, 12, 3671-3691, https://doi.org/10.5194/tc-12-3671-2018, 2018. So this limit should be clearly mentioned in the text.

**Reply:** We add a reference about the comparison between SIT from ORAS5 and from L-band observations (SMOS-SIT) for thin Arctic sea ice in the revised manuscript (Line 125, as reply to RC#1-7).
"*In comparison with other ocean-sea ice reanalysis (Chevallier et al., 2017), ORAS5 represents Arctic sea ice reasonably well and the errors in SIV (up to 10%) are comparable to the uncertainties in PIOMASS (Schweiger et al., 2011). Tietsche et al. (2018) found there is good agreement between SIT from ORAS5 and from L-band observations for thin ice in the freezing season (October–December) with respect to the interannual variability and trends of thin sea-ice area over the pan-Arctic. Therefore, it is a reasonable choice to apply ORAS5-SIT to initialize the decadal prediction, which typically starts on 1 November.*"

To support this statement, we compare the anomalies of November mean NH SIE/SIV between ORAS5 (5-member) and OBS (NSIDC/PIOMASS) in Fig. X (see below). Figure X shows that REF and OBS generally agree well during the study period 1987-2016.

We add a statement about the limit of this study by using ORAS5-SIT in the last section (Line 636). " *A limit of SIT from ORAS5 is that has no constraint by SIT observations. Although we only initialize SIT in November, in which ORAS5-SIT has shown some reliable results in representing thin ice during the freezing season (October–December) over the pan-Arctic when compared with observations by Tietsche et al. (2018). Xie et al. (2018) by directly assimilating SIT show that the perennial (thick) ice in CAO could be remarkably corrected if compared with the SIC assimilation. Therefore, the present study may underestimate the impact of SIT initialization to CAO-MYI on the Arctic climate prediction skill and further investigation is needed for an accurate assessment.* "

[Figure]

Figure X. Same as Fig.3, but anomalies of November mean NH SIE/SIV (w.r.t. the climatology over 1979-2018), compared between ORAS5 (5-member) and OBS (NSIDC/PIOMASS).

3. Line 98-103: A 25-member ensemble has been generated by the EC-Earth3, but this study only uses the 5 members of them. What reasons has been used to choose the rest 4 members excluding the FREE1? If no direct relation between these 25 members and the used 5 members, the concerned statements may be redundant for the readers.

**Reply:** It is because there are five members of the initialized hindcast simulations, we select additional 4 members besides FREE1 from the 25 members uninitialized simulations to comprise a 5 member ensemble (so-called FREE) in order to perform a fair assessment of the forecast skill. We think it is necessary to mention the 25 members, because the members of FREE are selected with consideration to represent well the overall feature (mean and variability) of the full ensemble (Fig. S1b, pink for FREE and black for full ensemble). We add explanation to Line 108-111.

Additional information was given in the original figure caption of Fig. S1 "*The 5 members are selected to represent historical simulations stated from different state in the time series of the global mean TAS, i.e., r1 - on an "average" state, r4 and r5 - on a state of relatively cold TAS, and r8 and r18 on a relatively warm state.*"

4. In these sensitive experiments, the derived anomaly for initialization is quite important. So as my understanding from Table 2, the initialization anomaly for sea ice on average is equivalent to the model biases shown in Fig. 2. Could you give some comments about their differences. furthermore, if comparing the stippled SIT in Fig. 2b and in Fig. 6, they look different, is it true and the reasons?

**Reply:** We partly agree. The initialized anomalies on November 1 are calculated as the difference between the REF state (in 20 years 1997-2016, or backwards to 1987-2006) and the mean state of REF (over 1979-2014). Therefore, the average of anomalies (SIE or SIV) over one 20-year period in this study can be negative, but not equal to zero.

The anomaly-corrected initial state (IC) is equivalent to the reanalysis state (REF) plus bias, according to the following expressions:
$Bias = MOD^{mean} - REF^{mean}$,
$Anom^{day} = REF^{day} - REF^{mean}$
$IC^{day} = Anom^{day} + MOD^{mean} = (REF^{day} - REF^{mean}) + MOD^{mean} = REF^{day} + (MOD^{mean} - REF^{mean}) = REF^{day} + Bias$

In anomaly initialization, there are few adjustments should be made, which can result in inconsistency with the REF anomaly. For example, there are warm model bias in sea ice states, which means $MOD^{mean} = 0$, $REF^{mean} > 0$. If in some years $Anom^{day} < 0$, we have to ignore the negative anomaly of REF, and set $IC^{day}$ to 0 (see Line 142-144). We note that the model biases are not removed from the initial states, and the model biases (i.e. system errors) have seasonal cycles and vary with regions.

We emphasize the difference between REF anomaly and model bias and how they contribute to forecast errors in Section 3 (from Line 300. As reply to RC#1-16.3.)
"*Figure 3 depicts different sea ice initialization strategies, e.g. in the first year low-SIC (FREE1) and high-SIT (FREE1) for AI0 in contrast to high-SIC (REF) and low-SIT (REF) for AI2. AI1 has the combination of high-SIC (REF) and high-SIT (FREE1). We note that AI0 and AI1 are initialized with different combinations of ocean (REF) and sea ice anomalies (REF or FREE1) while A2 is not. As model physics differ, initialization*

*shocks likely impact the prediction skill. In the meantime system errors (not corrected in anomaly initialization) may develop fast, if the REF anomalies of FYI (<10 % SIC and 0. 2 m SIT) are much smaller than the positive (negative) bias (>20 % SIC and >1 m SIT), such as in the Atlantic (Pacific) sector in Fig.S2. Therefore, it is essential to track the development of initialization shocks and system errors in the next section."*

We characterize the range of anomaly in the CAO and MIZ in Line 342.
*" When we calculate the anomaly of regional averages from REF (i.e. ORAS5), the Arctic Ocean is fully covered by sea ice during winter and there are neither trends nor interannual variability in SIE (SIC changes below 5%) and year-to-year changes in MYI-SIT up to 1 m. One exception is the Barents Sea with high variability in SIC because it is open to the Atlantic inflow. In the Atlantic/Pacific MIZ, the warm Atlantic/Pacific waters regulate changes in the very thin FYI states (below 10% SIC and 0.2 m SIT)."*

We confirm that the stippled areas for SIT in Fig.2b look identical to Fig.6 if we apply identical marker and bordering latitude. We add it to the last sentence in Fig.6 caption.
*" The stippled areas in the middle columns are the same as in Fig. 2b."*

[Figure]

Fig.2b                                        Fig.6

5. Fig. 2b shows the SIT with dots covers a considerable area in the Arctic. Does it necessary to skip so wide area like in the Laptev Sea and the East Siberian Sea, or give more explanation for the threshold of 100 m? (Line170)

**Reply:** Yes. There are several reasons: 1) observation uncertainties are quite high in the Laptev/East Siberian Seas. It is very common to exclude those regions in the analysis (such as Fig.2b in Xie 2018, Fig. 1 in Kwok, 2018) or nudge observational SIT only if SIC>40% (Blockley and Peterson, 2018); 2) The regions are covered by FYI (Tilling et al, 2018), which is indicated by the fact that they are within the September sea ice extent climatology. The added skill of initialized local SIT from November will not last over the melting season of the first forecast year. Our interest is the impact of MYI-SIT in the CAO on decadal time scale; 3) technically, the ORAS5-SIT anomaly is regridded from 0.25 to 1 degree horizontally for initialization without spatial smoothing. Therefore, in some years, there are few isolated grid-points with thin ice or ice holes in the dotted regions, and thus the ice is highly dynamic. It results in extremely high water levels and wind speeds which crash the model in a few time-steps. Therefore, we introduce a mask of coastal water, gradually from 0.5m to 100 m depth to avoid correcting ORAS5-SIT anomalies. In the dotted region, ORAS5-SIC and FREE1-SIT are used in AI2, same as AI1.

We add more explanation in Section 2.1 (Line 197-203).
*"We note that the dotted regions cover a considerable area in the Laptev/East Siberian Seas (Fig. 2b), where SIT observation uncertainties are quite high and are typically excluded from the analysis (Kwok, 2018; Xie et al., 2018). Furthermore, because the regions are highly dynamic and covered by first year ice (FYI, Tilling et al., 2018), the added skill of local initialized SIT is not expected to last over the melting season. In order to identify the added skill from MYI-SIT in the CAO to its adjacent waters on decadal time scales, the ORAS5-SIC anomaly and FREE1-SIT ICs are implemented to the dotted regions in AI2 as in AI1. In this way, the skill difference between AI2 and AI1 in years 2-9 should arise from a remote (MYI) origin."*

6. Line 314: ". . ., RMSESS>0.2 is commonly found poleward of the September sea ice edge determined by FREE (Fig. 2a), . . ." Clearly, this inferred statement lack of enough proofs to attribute to the misfits in

September alone.

**Reply:** We removed this statement and the assessments have been restructured because of new defined regions from geographical to physical regimes (as reply to RC#1-1.2). We assess two domains: CAO and MIZ (from Line 378 and L389, respectively).

7. Figure 7 show the 95% confidence lines which is quite interesting for me. Could you introduce some words how to evaluate these results? If possible, add one paragraph for this methodology in text.

**Reply:** as reply to RC#1-16.5. We make the following changes:
We introduce how to evaluate these results in Line 420:"*Additionally, the thin lines in Fig. 7 represent the upper/lower bounds of the 95 % confidence intervals obtained with a t-distribution for correlations and a χ2 distribution for RMSE. The correlation with one experiment is not significant, if the confidence interval goes below 0. Furthermore, the difference between two experiments is not significant, if those two intervals overlap."*

The results in section 3.3.2 are rewritten, by adding how to interpret the 95% confidence lines in order to evaluate the skills of initialized experiments.

For ACC (TAS) in Fig.7c, stippled points are added if correlation is insignificant ($p >= 0.05$).

Other small comments:
8. Does the TAS (near surface air temperature) mean the air temperature at 2 m?

**Reply:** "*TAS is a MIP variable, defined as air temperature at 2 m.*" We add it in Table 1 notation and introduce it in the revised manuscript (Line 63).

9. In Table 1, the reference dataset of ORAS5 has been signed 5 for the ensemble size. As we known, the ORAS5 reanalysis assimilated the observations in ocean and the satellite-based sea ice concentration. The related 5 members from ORAS5 are not clear, and especially for their differences.

**Reply:** ORAS5 is a global ocean-sea ice ensemble from the ECMWF. We add a short description about the 5 members and the ensemble spread in the revised manuscript (Line 118-120).
"*ORAS5 (See details in Zuo et al., 2019) is a global ocean-sea ice ensemble and its five ensemble members are used to account for observations uncertainties and analysis errors in the surface forcing. In this study period, the ensemble spread for sea ice is found to be representative for the analysis errors.*"

10. Line 304: replace "INIT denote experiments with different initialization methods, i.e. ..." with "INIT denotes the different experiment, i.e. . . . ."

**Reply:** We change it in Line 373.

11. Line 345: It said the polar cap domain (see Fig. 2a), but the caption of Fig. 5 also mentions the polar cap domain (north of 70N). To avoid the confuse, please keep the same concept in this study.

**Reply:** It was defined first in the original manuscript (Line 255). We have removed the repeated information to avoid confusion.

12. Checking the abbreviations are explained when they emerge at first time: ORCA at Line 94; ORAS4 at Line 106; SSP2-4.5 at Line 192;

**Reply:** We added them in the revised manuscript (Line 100, 115, 224).

**Mentioned literature:**

Armitage, T. W., Manucharyan, G. E., Petty, A. A., Kwok, R., and Thompson, A. F.: Enhanced eddy activity in the Beaufort Gyre in response to sea ice loss, Nat. Commun., 11, 1–8, 2020.

[revised manuscript text omitted]